# A novel DNA primase-helicase pair encoded by SCC*mec* elements

**Aleksandra Bebel†, Melissa A Walsh, Ignacio Mir-Sanchis‡, Phoebe A Rice***

Department of Biochemistry and Molecular Biology, University of Chicago, Chicago, United States

**Abstract** Mobile genetic elements (MGEs) are a rich source of new enzymes, and conversely, understanding the activities of MGE-encoded proteins can elucidate MGE function. Here, we biochemically characterize three proteins encoded by a conserved operon carried by the Staphylococcal Cassette Chromosome (SCC*mec*), an MGE that confers methicillin resistance to *Staphylococcus aureus*, creating MRSA strains. The first of these proteins, CCPol, is an active A-family DNA polymerase. The middle protein, MP, binds tightly to CCPol and confers upon it the ability to synthesize DNA primers de novo. The CCPol-MP complex is therefore a unique primase-polymerase enzyme unrelated to either known primase family. The third protein, Cch2, is a 3'-to-5' helicase. Cch2 additionally binds specifically to a dsDNA sequence downstream of its gene that is also a preferred initiation site for priming by CCPol-MP. Taken together, our results suggest that this is a functional replication module for SCC*mec*.

**\*For correspondence:**
price@uchicago.edu

**Present address:** †Phage Consultants, Gdynia, Poland; ‡Umeå University, Umeå , Sweden

**Competing interests:** The authors declare that no competing interests exist.

## Introduction

*Staphylococcus aureus* is a dangerous human pathogen, due in part to the emergence of multi-drug-resistant strains such as MRSA (methicillin-resistant *S. aureus*). MRSA strains have acquired resistance to β-lactam antibiotics (including methicillin) mainly through horizontal gene transfer of a mobile genomic island called staphylococcal cassette chromosome (SCC) (*Moellering, 2012*). SCC*mec* is a variant of SCC that carries a methicillin resistance gene, *mecA*. SCC*mecs* have been found in all known MRSA isolates, highlighting the crucial role of the element in the emergence of MRSA. Despite its clear medical importance, very little is known about how SCC*mec* is propagated in bacteria, especially about the steps occurring between excision from the host genome and transfer to new bacterial host, which could include replication of the circularized element as previously shown for certain other mobile genomic islands (*Novick et al., 2010*). Determining the biochemical functions of the conserved proteins encoded by these elements can shed light on these processes and advance our understanding of SCC*mec* biology.

SCC elements show great diversity in size and gene content, with two common features: a single integration site within the *S. aureus* genome, and a conserved cassette chromosome recombinase (*ccr*) locus (*Ito et al., 1999*). To date, at least 14 distinct types of SCC*mec* have been classified based on the features of their *mecA* and *ccr* loci (*Baig et al., 2018*; *International Working Group on the Classification of Staphylococcal Cassette Chromosome Elements (IWG-SCC), 2009*). In addition to the conserved ORFs, SCC elements may carry different cargo, such as ORFs providing mercury or fusidic acid resistance (*Ito et al., 2001*; *Lin et al., 2014*). SCCs also encode many so-far uncharacterized ORFs, presenting opportunity for discovery of proteins with interesting novel functions and potential biotechnological uses.

Recently, SCC elements have been divided into two groups based on the pattern (arrangement and predicted domains) of the conserved ORFs in their *ccr* loci (*Figure 1*; *Mir-Sanchis et al., 2016*). The first operon differs most between the two patterns. In pattern 2, it encodes three proteins, which are the focus of this work: a putative A-family DNA polymerase CCPol, MP (a small protein

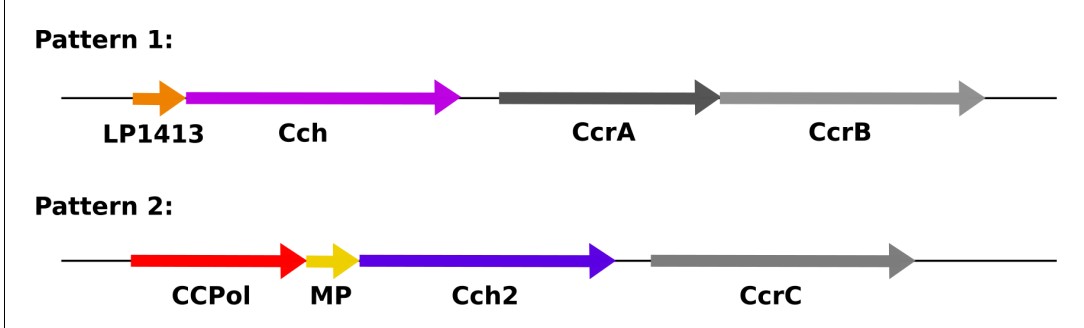

**Figure 1.** Organization of conserved ORFs in SCC elements. Pattern 1 elements (including types I-IV, VI, and VIII-XI) carry an operon encoding a short ssDNA-binding protein, LP1413, and a helicase, Cch, whereas Pattern 2 elements (including types V, VII, and XII-XIV, and SCC*mer*) carry an operon encoding CCPol, MP, and Cch2, which are characterized in this manuscript. *Figure 1—figure supplement 1* shows the sequence alignments of CCPol, MP, and Cch2 from various SCC elements. The recombinase genes (CcrAB or CcrC) are found after these operons, followed by three additional small ORFs not shown.

The online version of this article includes the following figure supplement(s) for figure 1:

**Figure supplement 1.** Sequence alignments of Cch2 operon proteins CCPol, MP, and Cch2.

with no conserved domains), and a putative helicase Cch2 (*Figure 1—figure supplement 1*). In pattern 1, the operon encodes a small single-stranded DNA-binding protein, LP1413, and a different helicase, Cch, that shows close similarities to the archaeal/eukaryotic MCM replicative helicase (*Mir-Sanchis et al., 2016*; *Mir-Sanchis et al., 2018*). Central to both patterns are the site-specific recombinases - CcrA and CcrB (pattern 1) or CcrC (pattern 2) - that excise SCC from the host and integrate it into the recipient genome (*Ito et al., 1999*; *Misiura et al., 2013*). Additionally, both patterns encode three small ORFs (not shown in *Figure 1*) containing domains of unknown function (DUFs) 950, 960, and 1643. The first has been renamed SaUGI, as it acts as a staphylococcal uracil-DNA glycosylase inhibitor, and we propose that it could protect newly synthesised SCC DNA from host UDGases as seen in phage replication (*Serrano-Heras et al., 2008*; *Wang et al., 2014*). The presence of a repertoire of proteins consistent with replication (replicative helicase, ssDNA-binding protein, putative polymerase, and SaUGI) led to the idea that SCCs might replicate after excision (*Mir-Sanchis et al., 2016*). Replication could increase the copy number of the element upon excision therefore assuring its stable inheritance as well as more efficient transfer to a new host strain. It could also increase the copy number of cargo genes when under selective pressure (*Gallagher et al., 2017*).

Interestingly, the helicase-containing operons of patterns 1 and 2 seem to have evolved separately and do not share homology (*Mir-Sanchis et al., 2016*). Cch and Cch2 belong to different families of helicases; each is predicted to be a distant homologue of a different self-loading replication initiator (Rep) helicase from unrelated staphylococcal genomic elements, the Staphylococcal Pathogenicity Islands (SaPIs) (*Mir-Sanchis et al., 2016*). Cch, containing DUF927, is related to SaPIbovI Rep helicase while Cch2 contains a Primase_Cterm domain similar to that of SaPI5 Rep, which is also found in many virus and plasmid primase-helicase fusion proteins (these are often also annotated as containing a D5_N domain, referring to the N-terminal portion of the helicase segment of the vaccinia virus D5 primase-helicase fusion protein) (*Hutin et al., 2016*). The two small proteins, LP1413 and MP share no homology; and the putative polymerase CCPol (previously named PolA for its similarity to A family polymerases) is only present in pattern 2 SCCs. Therefore, the structural and functional understanding of pattern 1 proteins gathered to date likely does not fully recapitulate the role of pattern 2 proteins.

Many mobile genetic elements (MGEs) that undergo replication, such as pathogenicity islands, plasmids, and viruses (including phages), encode an initiator helicase, a primase (occasionally both proteins are fused), and a recognized origin of replication. A notable example are the aforementioned SaPIs. SaPI replication requires a Rep helicase that uses its C-terminal domain to recognize an origin of replication, found directly downstream from the helicase gene and consisting of multiple short sequence repeats ('iterons') (*Ubeda et al., 2007*). Functionally, this is reminiscent of certain

viral helicases such as SV 40 large T antigen and papilloma virus E1 protein which also recognize their cognate origins of replication but do so using an N-terminal dsDNA binding domain (*Bergvall et al., 2013*; *Fanning and Knippers, 1992*).

Interestingly, SaPIs also encode a primase that plays a role in but is not essential for SaPI replication in vivo (*Ubeda et al., 2008*). Known primases belong to one of the two evolutionarily unrelated families: DnaG-like (bacterial primases containing conserved a TOPRIM domain *Aravind et al., 1998*) and archaeo-eukaryotic primases (AEPs; containing a highly derived version of the RNA recognition motif also found in certain RNA and DNA polymerases *Iyer et al., 2005*). Both primase types are employed frequently by MGEs, including elements integrated into the host genomes, with both the SaPIbov1 and SaPI5 primases belonging to the AEP family. Generally, most known primases synthesize short RNA primers with notable exceptions being human PrimPol (*García-Gómez et al., 2013*), a deep-sea vent phage NrS-1 PrimPol (*Guo et al., 2019*), or PolpTN2 from pTN2-like plasmids in Thermococcales (*Gill et al., 2014*), all of which prefer to synthesize DNA, and some of which make much longer products. While most often the RNA primers used for DNA replication are synthesized by specialized primases, there are instances where the host RNA polymerase has been co-opted to perform this function (*Kramer et al., 1997*; *Zenkin et al., 2006*). SCC elements encode proteins that could take on the canonical functions required for MGE replication: Cch and Cch2 could act as initiator helicases equivalent to the Rep proteins of the SaPIs, while the polymerase in pattern 2 SCCs could replicate the DNA. However, the other typical components of MGE replication systems – a canonical AEP- or DnaG-like primase and the origin of replication – were not previously identified for SCCs.

In this project, we biochemically characterized pattern 2 conserved proteins CCPol, MP, and Cch2. We showed that all three interact with each other in vitro. Cch2 is an active 3'-to-5' helicase that acts specifically on a putative SCC*mec* origin of replication sequence downstream of the operon. CCPol and MP form a complex that not only extends primers in a template-dependent fashion, but can also synthesize new DNA strands in the absence of primers. This DNA primase activity depends on the presence of MP, suggesting a novel primase-polymerase mechanism, and the preferred primase start site maps to the putative SCC*mec* origin of replication. The individual functions of the three proteins are consistent with their joint role in replication of SCC*mec*, where Cch2 would act as the replication initiator helicase and CCPol-MP would create DNA primers using the unwound origin as a template.

## Results

### SCCPol, MP, and Cch2 interact with one another

In pattern 2, the three ORFs (CCPol, MP, and Cch2) upstream from the recombinase gene were predicted to comprise a single operon, which we will refer to as the 'Cch2 operon'. We cloned the type V operon from the composite SCC*mec* element found in *S. aureus* strain TSGH17 (additionally carrying an SCC*mer*-type operon, with approximately 70% amino acid sequence identity between the two operons; see sequences in *Figure 1—figure supplement 1*). Operons identical to the cloned type V operon are found in SCC elements not only in other *S. aureus* strains, but also in *S. epidermidis*, *S. heamolyticus*, and other Staphylococci. To see whether all three proteins were expressed as predicted (e.g. without fusion by ribosomal frameshifting), we cloned the whole Cch2 operon into an *E. coli* expression vector with the first ORF (CCPol) His$_6$-tagged. Upon induction, we observed expression of all three proteins (*Figure 2a*). Furthermore, all three proteins co-eluted from nickel beads and from a size-exclusion column, suggesting that they interact with one another. We then performed a pairwise analysis: one of the proteins was His$_6$-tagged and co-expressed in *E. coli* with a second, untagged protein (*Figure 2b* and *Figure 2—figure supplement 1*). We observed interactions within all possible pairs (CCPol-Cch2, CCPol-MP, and MP-Cch2), with the strongest interactions between CCPol/MP and CCPol/Cch2 and the weakest interactions of tagged Cch2 with either CCPol or MP (suggesting that the N-terminal tag might interfere with the interactions). This indicates that all three predicted ORFs of the Cch2 operon are expressed as predicted and interact with one another.

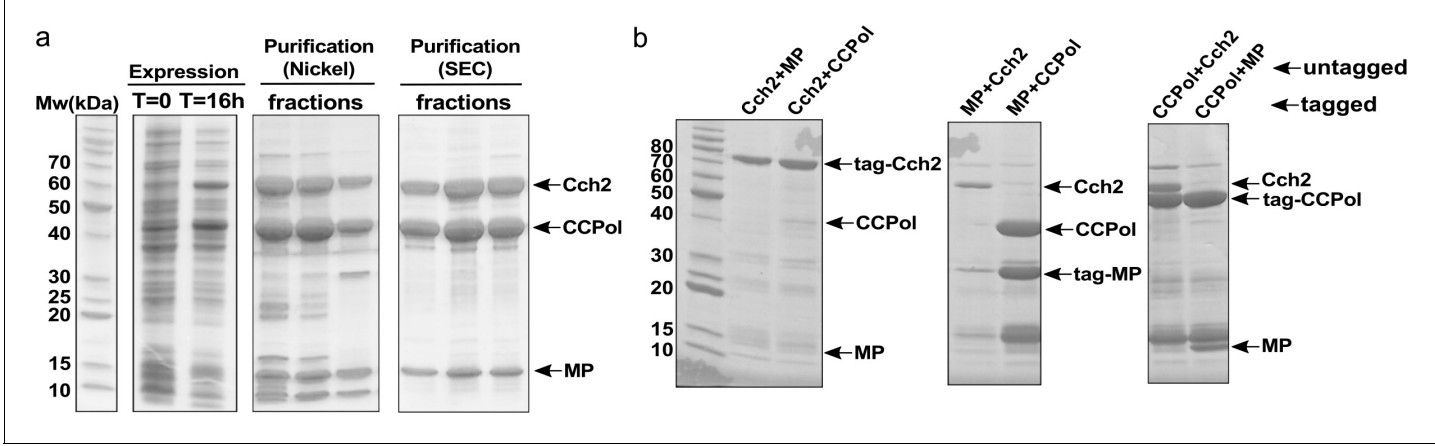

**Figure 2.** CCPol, MP, and Cch2 interact with one another. (a) Co-expression and co-purification of His$_6$-CCPol, MP, and Cch2 from the Cch2 operon of a type V SCC*mec* element; relevant fractions run on SDS-PAGE. *Figure 2—figure supplement 2* shows the oligomeric states of individually purified Cch2 and CCPol-MP. (b) Pairwise pulldowns of Cch2 operon proteins. One protein within each pair was tagged with N-terminal His$_6$-SUMO tag and co-expressed with a second, untagged protein. Cleared *E. coli* lysates were applied to Nickel-sepharose beads and eluted proteins were analyzed by SDS-PAGE. The contaminating band just above MP is most likely prematurely cleaved His$_6$-SUMO tag. See *Figure 2—figure supplement 1* for untagged controls.

The online version of this article includes the following figure supplement(s) for figure 2:

**Figure supplement 1.** Pairwise pulldowns of Cch2 operon proteins: untagged controls.

**Figure supplement 2.** Oligomeric states of purified Cch2 (a) and CCPol-MP (b,c).

## Cch2 forms oligomers and binds specifically to a putative origin of replication

To investigate the functions of the proteins encoded by the Cch2 operon, we started with characterization of purified Cch2. First, we investigated its oligomeric state on a blue native gel and found that it formed large multimers consistent with hexamers and dodecamers (*Figure 2—figure supplement 2a*), similar to Cch and replicative helicases such as MCM or DnaB (*Bailey et al., 2007*; *Miller et al., 2014*; *Mir-Sanchis et al., 2016*).

Next, we asked if Cch2 binds to specific sequences in dsDNA, based on analogy to SaPI Rep helicases that bind specifically to their cognate origins. SaPI origins are found just downstream from their *rep* genes (*Ubeda et al., 2007*), and contain a series of short repeats of a specific sequence (referred to as 'iterons') which Rep appears to bind as two hexamers. We therefore proposed that the intergenic region between the *cch2* and *ccrC* genes could act as an SCC*mec* origin of replication. This region is highly similar among different pattern 2 conserved loci and it contains indirect and direct repeats that could potentially serve as iterons (*Figure 3a*). Purified Cch2 formed two discrete complexes with dsDNA PCR products representing this intergenic sequence (*Figure 3b*, WT). Deletion of the inverted repeats did not affect binding (*Figure 3b*, -IR) while deletion of the direct repeats completely abolished it (*Figure 3b*, -DR). Furthermore, Cch2 did not bind to random DNA sequences with length and GC-content similar to the intergenic sequence (*Figure 3b*, R1and R2) unless the 23-bp direct repeat region was present (*Figure 3b*, R1+DR and R2+DR). To better assess affinity, assays were repeated with lower protein and DNA concentrations and short synthetic duplexes (*Figure 3c* and *Figure 3—figure supplement 1a*). While smearing and aggregation in the well under these conditions made quantitation somewhat subjective, binding was sequence-specific and the apparent Kd for the direct repeat-carrying duplex was between 100 and 200 nM. Binding to ss DNA was not detectable (*Figure 3—figure supplement 2*). We suspect that this was because our oligos had no blocking moieties to prevent it from sliding off the ends, since in order to perform its function a helicase must interact with ssDNA.

We predicted that Cch2 might have a similar domain organization to Cch or the SaPI Rep proteins, with the C-terminal domain-binding dsDNA and the core domain necessary for helicase activity. The isolated C-terminal domain of Cch2 (residues 400–538) bound specifically to the putative

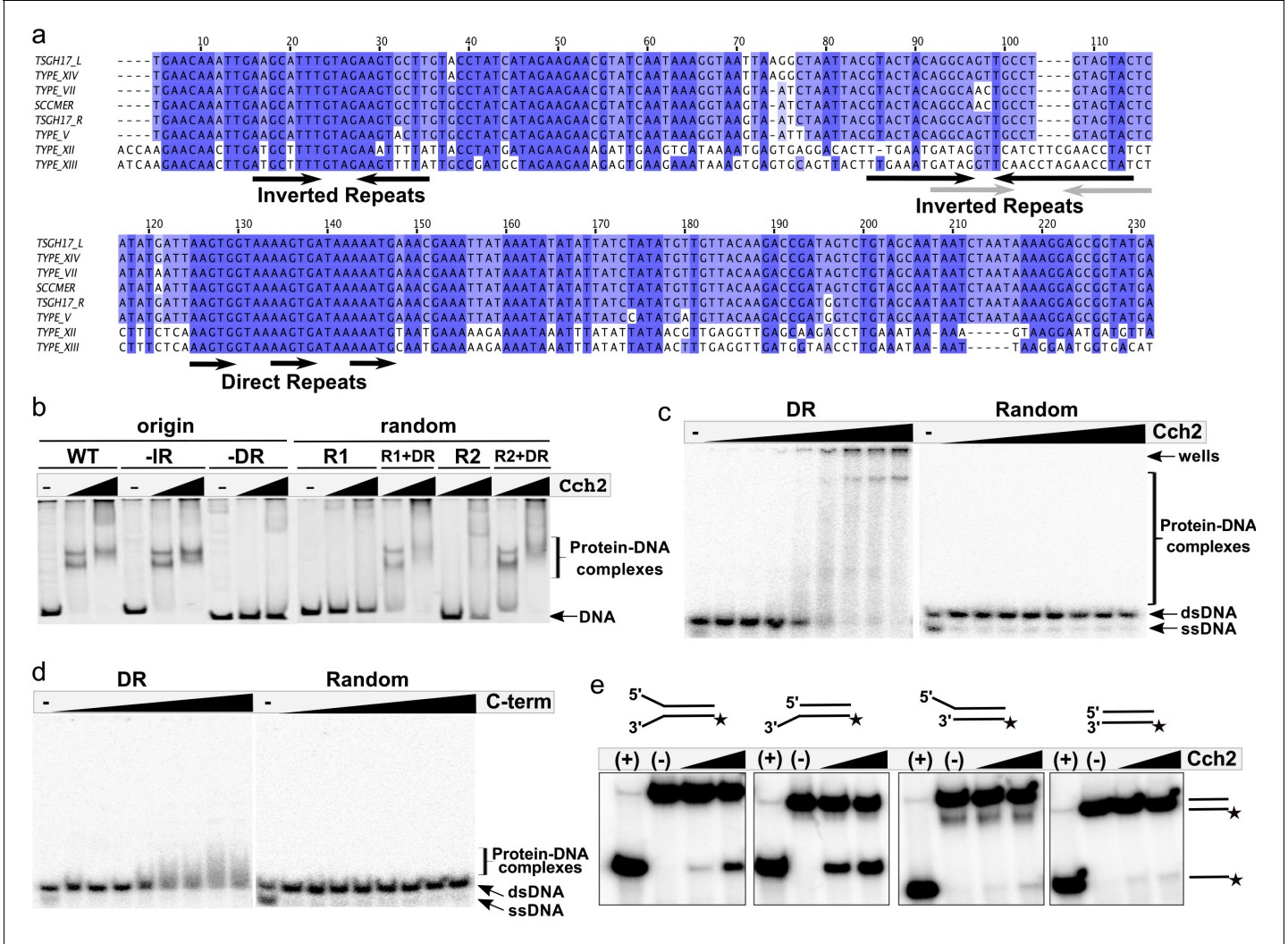

**Figure 3.** Cch2 binds a specific sequence downstream from its gene and is an active 3'-to-5' helicase. (a) Sequence alignment of the intergenic region between the *cch2* and *ccrC* genes, with eight sequences from different *S. aureus* pattern 2 SCC elements: the two gene clusters (L and R) from a composite SCC*mec* from TSGH17 strain (SCC accession number AB512767), type V element from WIS strain (AB121219), type VII from JCSC6082 (AB373032), type XII from strain BA01611 (JCSC6082), type XIII from isolate 55-99-44 (MG674089), type XIV from strain SC792 (LC424989), and SSC*mer* (carrying a mercury resistance cluster) from strain 85/2082 (AB037671). Shading indicates sequence conservation. Inverted repeats (IR) that could potentially form hairpins and direct repeats (DR) that could act as iterons are marked below the sequence; arrows in grey show inverted repeats found in type XII and XIII elements. (b) Cch2 binding to the PCR-amplified putative origin of replication, its derivatives, and synthetic duplexes of random sequence (R1 and R2), without and with (+DR) the direct repeat sequences. Final protein concentrations were 1 and 2 µM in monomers. (c) Cch2 (10–500 nM) binding to 23 bp ds-DNA oligos (5 nM) representing the DR from the putative origin of replication or a random sequence. The experiment was repeated and the results are shown in *Figure 3—figure supplement 1a*. (d) Binding of the Cch2 C-terminal domain (C-term (10–500 nM)) to 23 bp ds-DNA oligos (5 nM) representing the DR from the putative origin of replication or a random sequence. The experiment was repeated and the results are shown in *Figure 3—figure supplement 1b*. (e) Helicase assay with four different 60 bp substrates (forked, 3'-overhang, 5'-overhang, and blunt-ended) and Cch2 (2 and 3 µM). The star represents the $^{32}$P on the bottom strand. DNA unwinding results in the formation of ssDNA product that can be detected by native PAGE. *Figure 3—figure supplement 3* shows additional requirements for Cch2 helicase activity.

The online version of this article includes the following figure supplement(s) for figure 3:

**Figure supplement 1.** Binding of Cch2 and its C-terminal domain (C-term) to 23 bp ds-DNA oligos representing the DR from the putative origin of replication or a random sequence – repeat of the experiment shown in (a) *Figure 3c* and (b) *Figure 3d*.

**Figure supplement 2.** Binding of Cch2 to 23-nt ss-DNA oligos representing the top and bottom strands of the DR from the putative origin of replication.

**Figure supplement 3.** Cch2 helicase activity requires ATP, the AAGTG iteron sequence in the substrate oligo, and catalytic Lys252.

origin of replication (*Figure 3d* and *Figure 3—figure supplement 1b*), suggesting that it constitutes the dsDNA-binding domain of Cch2.

## Cch2 is an active 3'-to-5' helicase requiring the AAGTG iteron sequence for activity

To test the helicase activity of Cch2, we employed a series of radiolabeled dsDNA substrates as described in *Mir-Sanchis et al., 2016*. Cch2 most efficiently unwound the substrates with an unpaired 3' overhang, suggesting a 3'-to-5' polarity of Cch2 translocation (*Figure 3e*), similar to that observed for Cch and MCM helicases. Cch2 translocation depends on the presence of hydrolysable ATP (no activity was observed with ATPɣS, AMP-PNP, or dNTPs) and of the iteron sequence AAGTG within the substrate, and requires the presence of the predicted Walker A catalytic residue K252 (*Figure 3—figure supplement 3*). The fairly weak activity suggests that additional binding sites or co-factors may be required. However, none of the variations to our assay tested so far have improved activity. Altogether, these data show that Cch2 specifically recognizes and unwinds DNA sequences immediately downstream of its own coding region, which most likely function as an origin of replication.

## CCPol and the CCPol-MP complex are active polymerases

We also characterized the other two proteins from the Cch2 operon, CCPol and MP. In our hands, MP overexpressed in *E. coli* was poorly soluble, but when co-expressed with CCPol the two proteins formed a tight, soluble complex (*Figure 2—figure supplement 2b and c*). CCPol could also be purified independently.

The polymerase activities of both the CCPol-MP complex and CCPol alone were assessed in a primer extension assay (*Figure 4a*). In the presence of dNTPs, both CCPol-MP and CCPol showed efficient extension of DNA (*Figure 4a*, lanes 4 and 10) and RNA primers (lanes 2 and 8), while neither extended primers efficiently in the presence of NTPs (lanes 1, 3, 7, and 9). The observed primer extension was template-dependent (lanes 5, 6, 11, and 12).

CCPol consistently extended radiolabeled DNA primers in the presence of various templates (*Figure 4b*, T1–T4; sequences in *Supplementary file 1* under 'primer extension assays'), with the most prominent extension product one nt longer than the template, similarly to other PolA-family polymerases such as T7 or Taq. Moreover, addition of nucleotides to the 3' terminus of an oligonucleotide substrate, in other words terminal deoxynucleotidyl transferase activity, could be observed in the absence of a primer, with a strong preference for dCTP, at least with this template. (*Figure 4c*; lanes 6 and 7 vs. 3–5). In the absence of MP, only a faint signal for a single addition was seen. In the presence of all 4 dNTPs and MP, approximately 20 nt were added, perhaps due to MP facilitating the pairing of some microhomology between the 3' end of one template and the middle of another (*Figure 4c* lanes 2 vs. 8). We would like to note that similar results can be seen for different templates and appear to be an inherent property of the CCPol-MP complex, extending beyond the typical terminal transferase activity.

Sequence and structural predictions suggest that CCPol is a member of the A family of polymerases. Based on predicted secondary structure alignments of CCPol and other PolA-family polymerases (*Figure 4—figure supplement 1*), we mutated two residues within the CCpol-MP complex, CCPol D169 and D324, that are predicted to bind catalytic $Mg^{++}$ ions. Neither mutant extended a radiolabeled DNA primer (*Figure 4d*), confirming that the fundamental mechanism of DNA synthesis by CCPol is consistent with that of other PolA family members.

## MP contributes ssDNA-binding activity to the CCPol-MP complex

To further characterize CCPol-MP and CCPol, we tested their DNA-binding properties. CCPol-MP bound tightly but without sequence preference to ssDNA and bound much more weakly to dsDNA: when fit to a simple binding model the apparent Kd was 15–22 nM for three different ssDNA sequences, and ~120 nM for dsDNA (*Figure 5a and c*, and *Figure 5—figure supplement 1a*) (for these calculations, Bmax was fixed at 100% to lower the number of variables). Interestingly, while the Pol active site was not required for ssDNA binding, MP was critical (*Figure 5b* and *Figure 5—figure supplement 1b*). Two shifted bands were seen with these 30-nt substrates, and slight but systematic deviations from the fitted curves imply possible cooperativity. As all four experiments

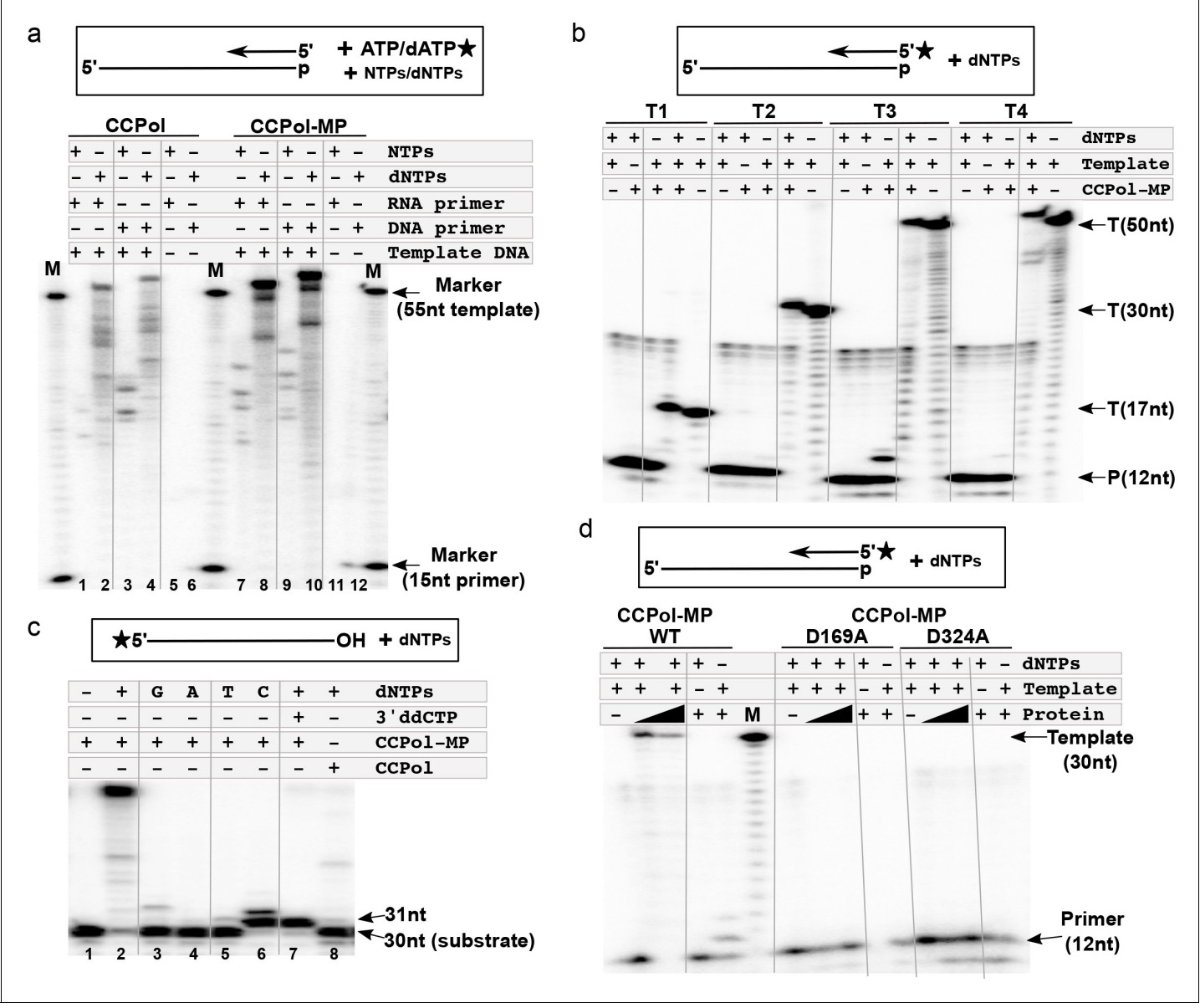

**Figure 4.** CCPol is an active polymerase. (a) Primer extension assay in the presence of CCPol-MP or CCPol (5 µM each), with DNA or RNA primers and dNTPs or NTPs. The 15-nt primer was extended in the presence of 55-nt complementary 3'-phosphorylated template and $\alpha$-$^{32}$P -labeled dATP/ATP as shown in the diagram; (b) Primer extension of a $^{32}$P-labeled DNA primer in the presence of CCPol-MP (5 µM), four different 3'-phosphorylated templates (T1-T4; sequences in *Supplementary file 1* under 'primer extension assays'), and dNTPs as indicated; (c) Terminal deoxynucleotidyl transferase activity of CCPol-MP and CCPol in the presence of individual ('G', 'A', 'T', and 'C') or combined ('+') dNTPs as indicated. As a control, template with 3' ddC (preventing addition of dNTPs) was used. (d) Primer extension assay with wild-type CCPol-MP complex and its variants containing predicted catalytic CCPol mutations (5 µM each) extending a $^{32}$P-labeled DNA primer in the presence of a 30-nt 3'-phosphorylated template and dNTPs as indicated. *Figure 4—figure supplement 1* shows structure-based sequence alignments of CCPol and other bacterial DNA polymerases. The online version of this article includes the following figure supplement(s) for figure 4:

**Figure supplement 1.** Structure-based sequence alignments of CCPol and the polymerase domains of several bacterial DNA Pol Is.

involving CCPol-MP and ssDNA yielded similar binding curves, we combined them to fit a Hill coefficient. This analysis gave Bmax = 88.6 +/- 2.3, Kd = 14.0 +/- 1.0, and h = 1.6 +/- 0.15, suggesting that binding might be partially cooperative. Finally, the pattern of shifted bands seen upon binding to a series of oligonucleotides of increasing length suggests that each CCPol-MP complex occludes approximately 15 nt (*Figure 5d*).

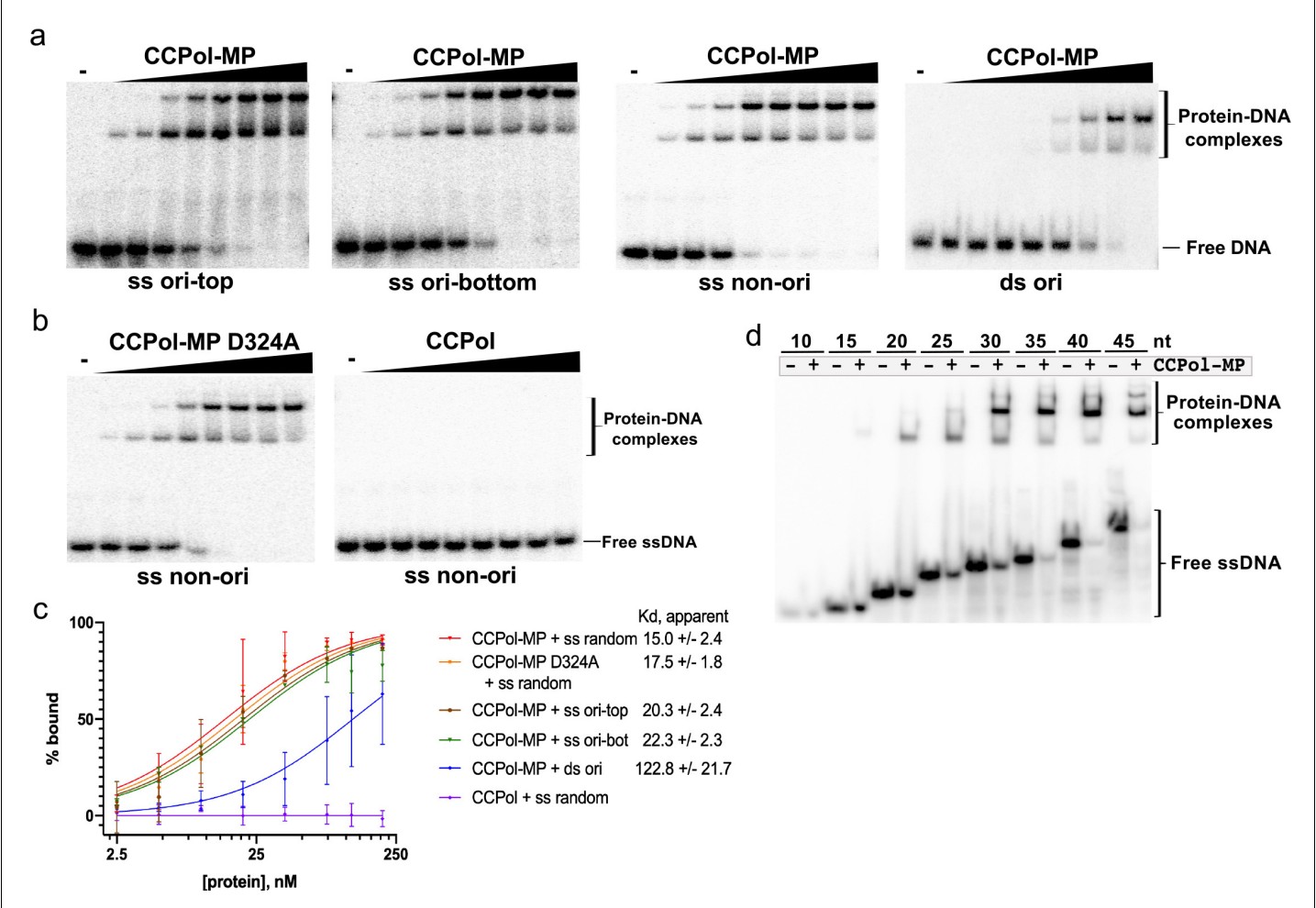

**Figure 5.** CCPol requires MP for ssDNA binding. (a) CCPol-MP (0–200 nM) binding to 30-nt ss and dsDNA substrates (2 nM) representing a fragment of the putative origin of replication (ori) or an unrelated sequence (non-ori) as determined by EMSA; (b) Binding to 30-nt ssDNA (2 nM) by CCPol-MP D324A and CCPol (concentrations of 0–200 nM). *Figure 5—figure supplement 1* shows the repeats of experiments in (a) and (b); (c) Quantification of DNA-binding assays shown in (a), (b), and *Figure 5—figure supplement 1*. Two repeats were quantified for each experiment; the mean and standard deviation are shown for each point. The curves show the fit to the data of the equation Y = Bmax*X/(Kd + X), where X equals percent bound, Y equals protein concentration, and Bmax is constrained to 100. For raw data, see *Figure 5—source data 1*. (d) CCPol-MP (180 nM) binding to ssDNA oligonucleotides ranging in size from 10 to 45 nt.

The online version of this article includes the following source data and figure supplement(s) for figure 5:

**Source data 1.** DNA binding data for CcPol-MP and CcPol.

**Figure supplement 1.** Binding of CCPol, CCPol-MP, and CCPol-MP D324A to 30-nt ss- and ds-DNA oligos representing a fragment of the putative origin of replication (ori) or an unrelated sequence (non-ori) – repeat of the experiments shown in (a) *Figure 5a* and (b) *Figure 5b*.

## CCPol-MP has primase activity

Since CCPol and MP are found within the same operon as the putative initiator helicase Cch2, we asked if they harbor primase activity despite carrying no identifiable primase motifs. To assess that, we used a non-denaturing gel to monitor the formation of a second DNA strand complementary to a radiolabeled template (*Figure 6a*). A slower migrating product could be detected for CCPol-MP both in the presence and absence of a primer (*Figure 6a*, lanes 3 and 5). Both observed products could be denatured by heat (lanes 4 and 6), showing that they are in fact forming a DNA duplex and are not a result of template extension (which is in principle blocked by the presence of a 3'dideoxy-nucleotide). This indicates that CCPol-MP can synthesize DNA de novo and therefore acts as a DNA primase. Less duplex product appears in the absence of primer suggesting that initiation of DNA synthesis is less efficient than simple elongation. Binding the first two dNTP's and/or forming the first

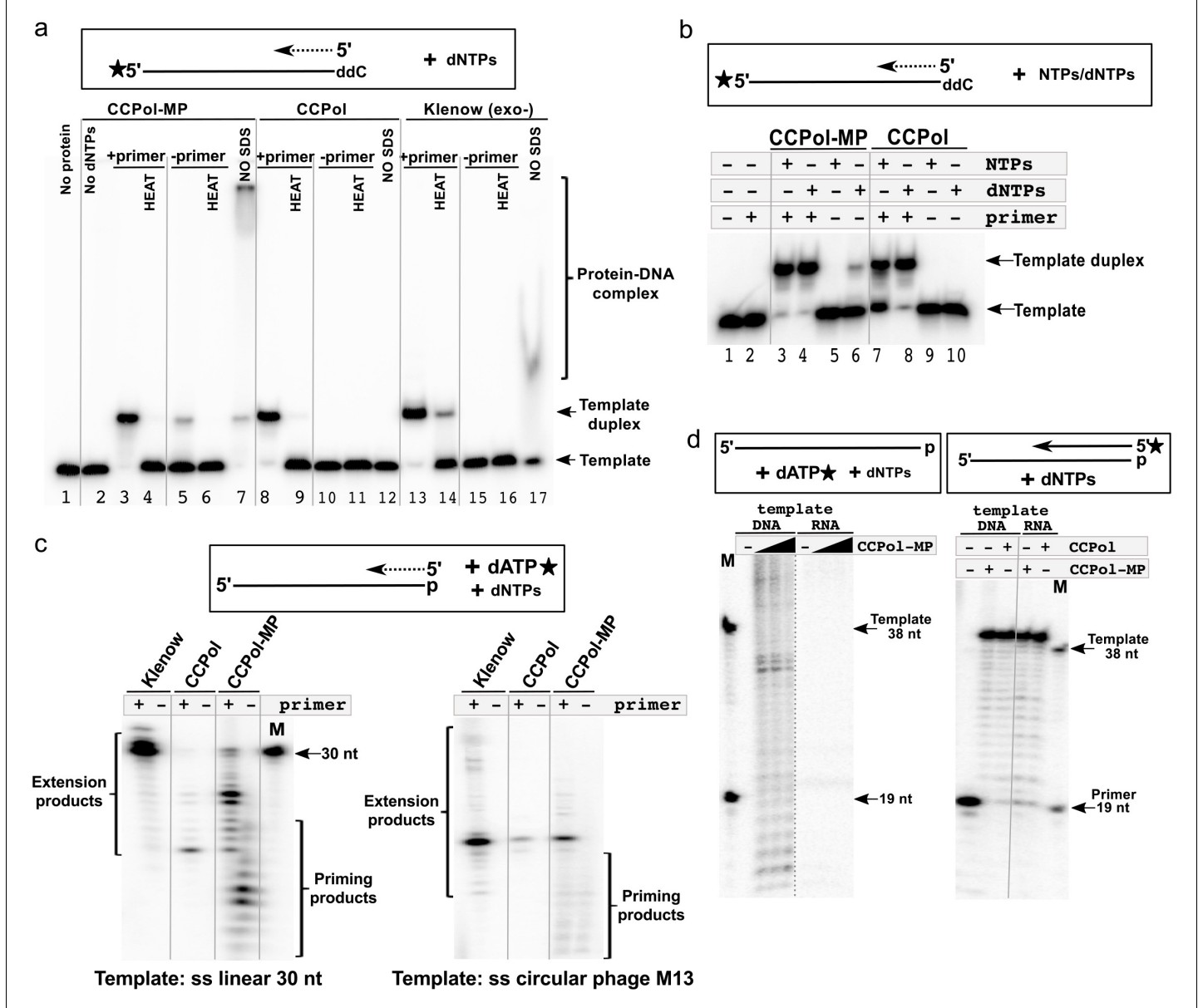

**Figure 6.** CCPol-MP has primase activity. (a) Primase assay with CCPol-MP, CCPol, and Klenow (exo-) polymerases. Formation of a template-product duplex was monitored on a non-denaturing gel in the presence or absence of a complementary primer (12 nt). The 3' end of the template (30 nt) was blocked with ddC. Formation of DNA duplexes was confirmed by denaturation back to ssDNA ('HEAT'). Samples without added SDS buffer ('NO SDS') show native DNA-protein complexes; (b) Primase/primer extension assay on a non-denaturing gel with CCPol-MP and CCPol (5 µM each) in the presence of radiolabeled 30-nt ssDNA template and dNTPs or NTPs as indicated; (c) Priming/primer extension assay with short (30 nt) linear (left) and phage M13 circular ssDNA (right) templates in the presence of α-32P dATP, in the presence or absence of a primer (12 and 24 nt, respectively); (d) Primase (left panel) and primer extension (right panel) assays in the presence of a DNA or RNA template and CCPol-MP or CCPol as indicated. M = markers with sizes corresponding to the primer (19 nt) and template (38 nt). The dashed grey line indicates that irrelevant lanes were removed from the gel; the solid grey lines were added to guide the eye.

chemical bond may be the rate- limiting step. Furthermore, since the duplexes observed both in the presence and absence of a primer run similarly in the presence of SDS, we conclude that the priming mechanism does not involve covalent protein-DNA interactions. Strikingly, in the case of CCPol alone duplex product could only be observed in the presence of a primer (lanes 8–11). As an additional control, we showed that the Klenow (exo-) fragment of *E. coli* DNA polymerase I, as expected, does not display primase activity (lanes 15 and 16) while it is efficient in primer extension (lanes 13

and 14). Finally, we looked at the protein-DNA complexes formed in this assay by omitting the SDS otherwise used to dislodge bound proteins. Consistent with previous results, CCPol-MP preferentially shifted the ss template DNA over the ds product (lane 7) while CCPol alone did not bind the template DNA (lane 12). Altogether, the results suggest that CCPol-MP is an active DNA primase, and that the MP subunit of the complex contributes the ability to synthesise DNA in the absence of primers.

## CCPol-MP requires dNTPs and a ssDNA template for primase activity

Having established that CCPol-MP is an active primase, we next use the duplex formation assay to characterize the requirements for efficient priming. CCPol-MP incorporated dNTPs much more efficiently than NTPs in our primer extension assays (see *Figure 4a*). In the priming assay, CCPol-MP could only form reaction products in the presence of dNTPs when no primer was added (*Figure 6b*, lanes 5 and 6), suggesting that CCPol-MP is a dNTP-incorporating primase. In comparison, CCPol alone was inactive in the absence of primers regardless of the nucleotides added (lanes 9 and 10), consistent with our proposal that MP is required for priming. In controls with a primer present, CCPol-MP and CCPol alone incorporated both NTPs and dNTPs.

In addition to the primase assays above that used a labeled template and monitored duplex formation, we confirmed these results with an assay directly monitoring formation of priming products on a denaturing gel by labeling the dATP to be incorporated by CCPol-MP. Consistent with our previous assay, only CCPol-MP and not CCPol alone or Klenow (exo-) showed DNA synthesis in the absence of a primer, while all three proteins could extend primers with varied efficiencies (*Figure 6c*, left panel). A similar pattern was observed when ss circular DNA (phage M13) was used as a template: priming products were observed only in the presence of CCPol-MP (*Figure 6c*, right panel). Note that the primase activity does not appear to have specificity for the moiety on the 3' end of the template, as in the duplex-forming assays the template 3' end was blocked a ddC moiety (*Figure 6a and b*) while in this dATP-incorporating assay it was blocked with a phosphate group. The primers seen in *Figure 6c,d* are relatively short, probably because the concentration of radiolabeled dATP (roughly 10 nM) used was limiting, but possibly also due to an intrinsic barrier to further elongation that is overcome at higher dNTP concentrations, as discussed in more detail below.

We were unable to detect any primase activity in the presence of a ss linear RNA template, while the matching DNA template enabled synthesis of DNA products (*Figure 6d*, left panel). At the same time, the RNA template did support primer extension, further characterizing CCPol-MP as having reverse transcriptase activity. Together, these results suggest that efficient CCPol-MP priming activity depends on the presence of a ssDNA template.

## The CCPol active site and divalent metal ions are required for primase activity

To test whether MP facilitates primer synthesis by the CCPol active site or utilizes a separate active site, we looked at the primase activity of CCPol-MP variants harboring the catalytic CCPol mutations described earlier, D169A and D324A. Both variants of CCPol-MP were as inactive as CCPol alone (*Figure 7a*), showing that primase activity, while facilitated by MP, requires the catalytic site of CCPol.

Next, we asked whether or not the subunits within a complex could exchange. CCPol and MP form a complex that is stable even at 1M salt concentrations (as determined by size exclusion chromatography and blue native PAGE, *Figure 2—figure supplement 2b and c*). However, addition of the WT CCPol to either inactive variant of CCPol-MP partially restored primase activity (*Figure 7a*), suggesting that the subunits within the complex are in a dynamic equilibrium and can interchange, or that they can act in trans between CCPol-MP complexes.

We also tested activity in the presence of different divalent metal ions (*Figure 7b and c*). Unsurprisingly, as they require the same active site, both primer extension and de novo synthesis were most efficient in $Mg^{++}$. $Ca^{++}$ did support primer extension although quite weakly at the lower protein concentration used. Perhaps because primase activity is generally weaker than primer extension, no priming products were seen in $Ca^{++}$. $Mn^{++}$ supported both activities but the products were shorter. We speculate that $Mn^{++}$ lowers the accuracy of nucleotide choice leading to mismatched primer ends that are poor substrates for extension and also cannot be edited because CCPol lacks

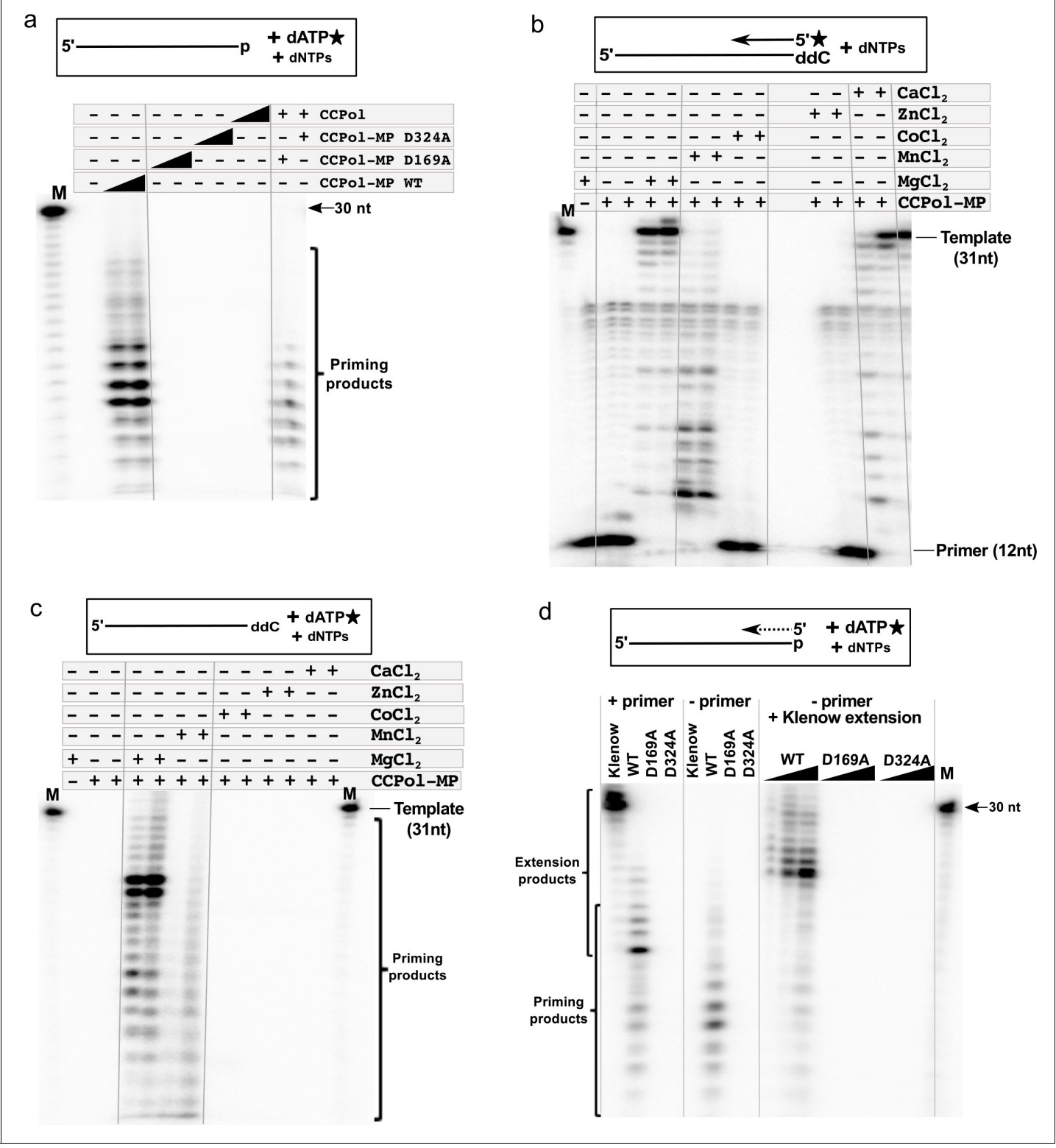

**Figure 7.** CCPol active site and divalent metal ions are required for primase activity. (**a**) Primase assay with CCPol-MP, CCPol-MP with the CCPol active site mutated, and CCPol alone (concentrations: 2 and 5 μM) in the presence of 30-nt ssDNA template and α-$^{32}$P dATP. The last two lanes show a complementation assay where CCPol-MP mutants (2.5 μM) were supplemented with WT CCPol (2.5 μM); (**b**) Primer extension of a $^{32}$P-labeled DNA primer (12 nt) in the presence of CCPol-MP (1 and 2 μM) and a 30-nt template with the 3' end blocked with ddC, in the presence or absence of selected divalent metal ions (20 mM); (**c**) Primase assay with CCPol-MP, a template (30 nt) with the 3' end blocked with ddC, and α-$^{32}$P dATP, in the

*Figure 7 continued on next page*

*Figure 7 continued*
presence or absence of selected divalent metal ions (20 mM); (d) Primase/primer extension with Klenow (exo-), CCPol-MP, and its two mutant variants
D169A and D324A (5 μM each in the left panel, 1–5 μM each in the right panel), with a 30-nt template and α-$^{32}$P dATP in the presence (left panel) or
absence (middle panel) of a primer. Some of the priming reactions were then further incubated with Klenow (exo-) to allow elongation of the CCPol-MP
– made primers (right panel). Unless otherwise noted, 5 μM CCPol-MP or CCPol and 5U Klenow were used in the experiments shown in this figure.

an editing exonuclease (*El-Deiry et al., 1984*; *Kunkel and Bebenek, 2000*). Neither Co$^{++}$ nor Zn$^{++}$ supported catalytic activity in either assay.

## CCPol-MP primers can be elongated by other polymerases

Our experiments showed that CCPol-MP can synthesise DNA in a primer-independent manner. At the same time, CCPol-MP polymerase activity is not as efficient as that of Klenow (exo-) polymerase (see *Figure 6c*). Direct comparison between the two polymerases shows that, under the conditions used, Klenow can extend the primer to the full length of the template while CCPol-MP extension products are shorter and less abundant, perhaps due to higher Km's for substrates and/or lower processivity, or perhaps due to an intrinsic preference for synthesizing short products (*Figure 7d*, left panel). At the same time, as seen before, Klenow cannot synthesise DNA de novo (*Figure 7d*, middle panel) – short primers have to be present for Klenow activity. CCPol-MP priming reactions further incubated with Klenow (exo-) resulted in appearance of products longer than those made by CCPol-MP alone (*Figure 7d*, right panel), suggesting that Klenow polymerase can extend primers made by CCPol-MP. Strikingly, Klenow extended most of the CCPol-MP-made primers to products seven nt shorter than the template, which suggests that the main primase starting site on this particular template is seven nt from the 3' end of the template (see more below). Moreover, we supplemented the inactive variants of CCPol-MP (D169A and D324A) with Klenow (exo-) and saw no priming or extension products, suggesting that Klenow cannot compensate for inactive CCPol in this assay, and therefore that, while MP can facilitate primer synthesis by CCPol, it cannot do so for a heterologous polymerase such as Klenow.

## CCPol-MP shows preference for the priming start site

When presented with templates of different, randomly chosen sequences but similar sizes, CCPol-MP began DNA synthesis in different places (or not at all), as seen in our prime-then-elongate assay (where after the priming reaction, additional unlabeled dATP is added to allow for further elongation of primers by CCPol-MP; *Figure 8a*). For example, a reaction in the presence of the 31-nt template 1 (2nd lane from left) and CCPol-MP yielded two major priming/elongation products of 23 and 24 nt. Considering that CCPol-MP tends to add one nucleotide beyond the template end (see *Figure 4b*), we can map the priming sites on that template to 7 and 8 nt downstream from the 3' end of the template. We performed such mapping for 25 different templates and identified 39 unique priming start sites (*Supplementary file 1*). Analysis of these priming start positions with respect to the 3' ends of their templates (*Figure 8b*) revealed that most start sites fell within 8–14 nt from the template end, with none closer than eight nt from the template end, which could suggest spatial constraints related to CCPol-MP binding.

We did not find a unique consensus start sequence shared by all of the templates apart from a strong preference for the first added deoxynucleotide to be a purine (in most cases, opposite a template cytosine), but a comparison of the 39 unique priming start sites allowed for a construction of a consensus priming start sequence logo (*Figure 8c*). This consensus sequence has vague but intriguing homology with the direct repeat region of the putative origin of replication 5' TACCACTTAAT (*Figure 8c* and see *Figure 3a*).

To further characterize the specificity of priming by CCPol-MP, the prime-then-elongate reactions were repeated with additional templates including the putative origin (*Figure 8d*). CCPol-MP did not prime efficiently on a control 30-nt poly-dT template, showing a faint smear of unspecific products. Replacing 8 of the 30 dTs with the origin sequence 5'ACCACTTA (dT-ori8-dT) made little difference other than shifting the distribution of faint products closer to the predicted start site

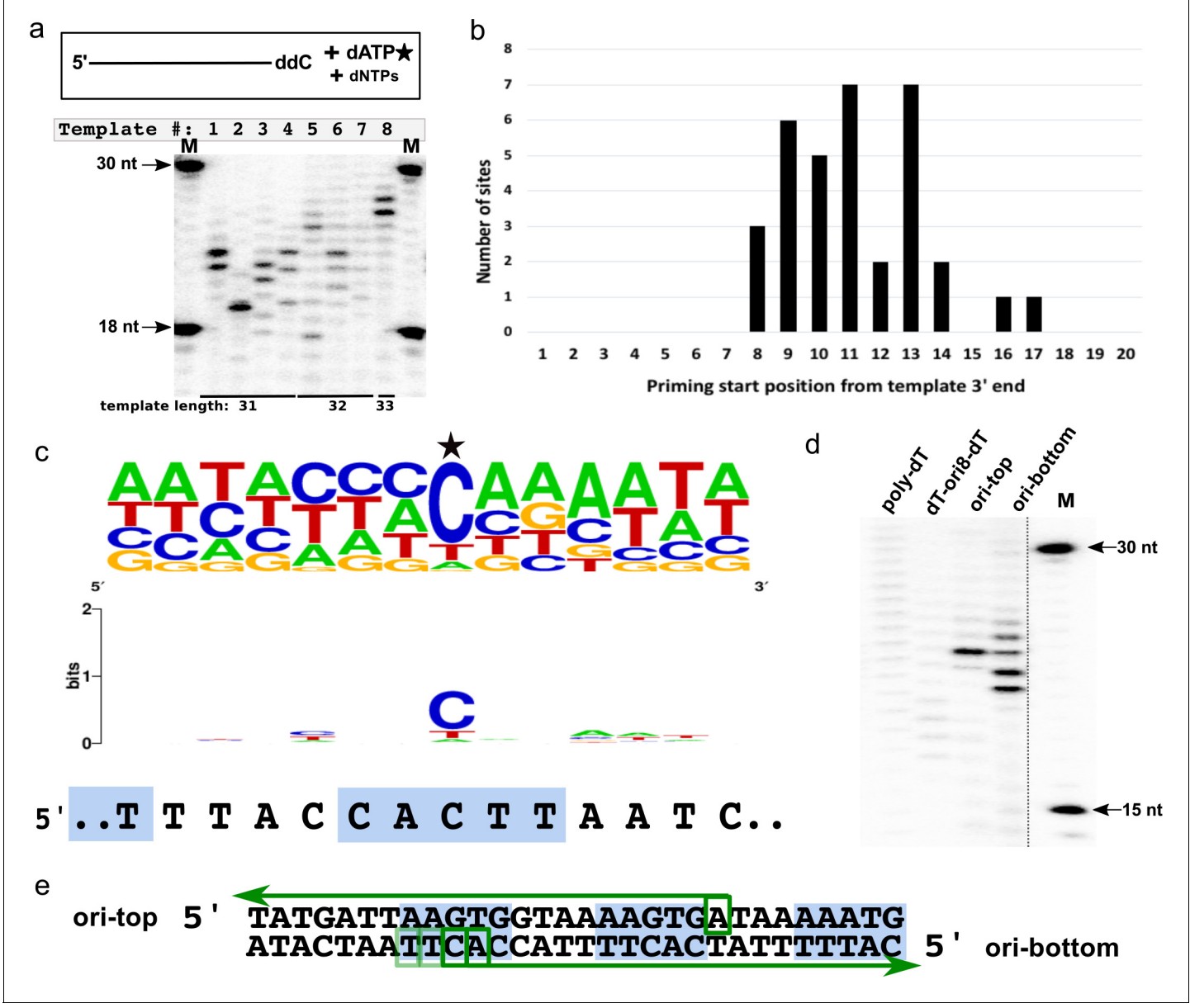

**Figure 8.** CCPol-MP priming start sequence preference. (a) CCPol-MP prime-then-elongate assay with eight different templates (31–33 nt including 3' ddC, as shown) with random sequences in the presence of radiolabeled dATP. Primers synthesized by CCPol-MP were further elongated by the complex upon addition of excess of dNTPs. Priming start sites can be mapped by reading the product size off the gel, then subtracting one nt to account for the terminal nucleotide added by CCPol-MP, and mapping the product length to the template starting at its 5' end; (b) Positions of 34 CCPol-MP priming start sites (*Supplementary file 1*) identified as in (a) with respect to the 3' end of the template; (c) Sequence consensus logo of the priming start sites identified as in (a). 39 unique priming start sites (*Supplementary file 1*) from 25 different templates are included in the logo, with the start site marked with a star. The sequence of direct repeat region within the putative origin of replication is aligned under the logo, with the 5-nt iterons marked by blue boxes; (d) CCPol-MP prime-then-elongate assay as in (a) with 30-nt templates (each with a 3' phosphate) as indicated; (e) Mapping of CCPol-MP priming start sites on the templates representing the putative origin of replication (ori-top and ori-bottom) based on the band position on the gel shown in (d). Green arrows represent the elongated DNA products, with green boxes marking the identified start sites. The darkness of the boxes corresponds to the efficiency of priming at that site as judged by the brightness of the bands in (d). Blue boxes mark the iterons within the putative origin of replication.

(*Figure 8d*), indicating the importance of the flanking sequences for efficient priming even in the presence of a favorable start site sequence. On the other hand, efficient priming was observed for both top and bottom strands of a template representing the direct repeat region of the putative origin of replication (ori in *Figure 8d*), with multiple products seen for the bottom strand (the strand

seen in *Figure 8c*) and mapping to the iteron seen in the start site sequence consensus (*Figure 8e*), confirming that this sequence is a valid CCPol-MP priming target. These results suggest functional interaction between the putative initiator helicase Cch2 and the primase/polymerase CCPol-MP.

## Discussion

In this work, we biochemically characterized three conserved proteins encoded by an operon found in all pattern 2 SCC elements. We find that these proteins physically interact with one another and have activities consistent with DNA replication: Cch2 is a 3'-to-5' helicase that binds specifically to sequence motifs found just 5' of its own coding sequence and CCPol is a DNA polymerase that can synthesize DNA primers de novo when complexed with MP.

The CCPol-MP complex constitutes a novel type of primase. Neither CCPol nor MP belong to either of the so-far characterized families of DNA primases: the DnaG-like or the AEP (archaeo-eukaryotic primase) families. In fact, NCBI annotates CCPol as carrying a 'PolA' conserved domain, and it has 15–22% sequence identity to other A-family polymerases. Based on modeling servers (*Yang et al., 2015*) and predicted secondary structure alignments (*Figure 4—figure supplement 1*), CCPol is closely related to bacterial DNA PolI enzymes, but contains only the polymerase domain and completely lacks the 3'-to-5' and 5'-to-3' exonuclease domains usually present in PolI enzymes. In our assays, CCPol without MP is a template- and primer-dependent DNA polymerase (*Figures 4a* and *6a*). However, upon addition of MP, the complex becomes proficient in template-dependent de novo synthesis of DNA (*Figure 6a and b*). We could not find any sequence- or secondary structure-based homologues of MP, nor could any known domains be identified. Furthermore, the largely beta predicted secondary structure of MP does not match the winged helix-turn-helix fold of the ssDNA-binding protein LP1413 from the pattern 1 SCC*mec* (*Mir-Sanchis et al., 2018*), despite its analogous position upstream from the Cch helicase gene.

A previous bioinformatic study included CCPol in a proposed new group of A-family polymerases termed the TV-Pols for their occurrence in transposons and viruses (*Iyer et al., 2008*). Due to their genetic association with D5-family helicases, which often have an N-terminal DnaG or AEP – type primase domain, it was suggested that the TV-Pols might act as primases. However, MP was not mentioned, and to our knowledge, none of the other members of the proposed TV-Pol family have been biochemically characterized.

The CCPol-MP primase-polymerase complex features a number of unique properties. As already mentioned, CCPol-MP is an exceptional DNA primase in that it cannot be classified into either of the two known families of primases. In contrast to most members of those families, CCPol-MP does not have a zinc-binding domain and does not appear to require zinc ions for activity (although it does require magnesium ions, consistent with the general metal-ion-dependent polymerase mechanism) (*Geibel et al., 2009*; *Kusakabe et al., 1999*; *Powers and Griep, 1999*). Furthermore, CCPol shows a strong preference for incorporating dNTPs over NTPs, and under our assay conditions CCPol-MP exclusively uses dNTPs to synthesize primers, which is striking since only a handful of known primases uses dNTPs rather than NTPs for primer synthesis (see Introduction). The structural features governing the choice of dNTPs over NTPs by a primase have been described for the AEP-like human PrimPol, where steric clashes between active site amino acids and the 2' hydroxyl of a ribose sugar prevent incorporation of NTPs (*Rechkoblit et al., 2016*). As CCPol is most closely related to DNA Pol I, we note that it does conserve two residues used to discriminate against ribonucleotides by *E. coli* PolI: E710 (our E174) and F762 (our F225).

Primases must be able to grip the template strand in the absence of a primer. In agreement with this, we found that MP confers tight ssDNA binding to the CCPol-MP complex even though binding to dsDNA was much weaker in our EMSAs (*Figures 5a* and *6a*). Other primases also utilize a second protein or domain to aid in binding the single-stranded template: DnaG and PrimPol use a zinc-binding domain whereas many AEP-family primases use a helical bundle that is sometimes fused to the catalytic module and sometimes part of a separate subunit (*Geibel et al., 2009*). Accessory domains and/or subunits have also been implicated in helping the primase enzyme bind the initial two nucleotide triphosphates, which MP might also do. These similarities suggest that in spite of involving different types of proteins, CCPol-MP might present yet another twist on a two-component primase system.

Clues about the CCPol-MP priming mechanism might be provided by the RNA polymerases of bacteriophages T7 and N4, which belong to the A family of polymerases yet can initiate synthesis of RNA de novo (*Cermakian et al., 1997*; *Lenneman and Rothman-Denes, 2015*; *Sousa, 1996*). These require an additional DNA promoter binding domain (*Dai and Rothman-Denes, 1998*; *Ikeda and Richardson, 1987*; *Kennedy et al., 2007*). CCPol-MP might function in an analogous way, with MP helping to deliver the single stranded template to the PolA-type active site. Another feature of these RNA polymerases is that the last helix of the thumb is shorter than in DNA Pol Is, allowing space for the triphosphate group of the first nucleotide. This helix is also predicted to be shorter in CCPol (*Figure 4—figure supplement 1*; residues 151–156 in CCPol vs. 684–692 for *E. coli* Pol I).

Primer synthesis by CCPol-MP frequently stopped at about 14-15nt in length (*Figures 6c* and *7a, d* and the shorter products in part c). These short products corresponded to assays that used a low concentration of radiolabeled dATP, which may have been limiting (further extension was observed when substantial additional dATP was added later in the reaction; *Figure 8*). However, the roughly consistent size of the primers made in *Figures 6c* and *7a,d* as well as the bimodal distribution of sizes in *Figure 7c* hints that MP acts as an 'anchor' through interactions with ssDNA upstream of the priming site, which would provide a barrier to elongation even if MP is flexibly linked to CCPol, and would limit the length of primers produced when the enzyme is in priming mode. High dNTP concentrations may force the system over that barrier, leading to longer products. Many other primases form products ranging from 5 to 10 nt, with a particular ability of most primases to count the exact length of a primer made (reviewed in *Frick and Richardson, 2001*; *Kuchta and Stengel, 2010*). For some primases, for example the human PrimPol, there is a proposed conformational change between the priming and elongation modes (*Bell, 2019*).

The elongation products that we observed suggested that CCPol-MP utilizes specific priming start sites. Within the consensus sequence for priming, the most commonly used nucleotides partially match (8/11 nts) a fragment of the putative origin of replication sequences, centering on one of its direct repeats (*Figure 8c*, top). Most primases do not start primer synthesis on a template at random but rather choose a specific site, with the level of specificity ranging from a simple requirement for a template pyrimidine (human primase) to a short (2–5 nt) specific template sequence (T4, T7 primases) (*Kuchta and Stengel, 2010*). CCPol-MP appears to start the primer synthesis at discrete positions, with the first nucleotide added most often being a dGTP (opposite to the template dCTP) and with 90% of the template start sites being a pyrimidine (*Figure 8c*). It is worth noting that T7 RNAP shows a similar bias towards using GTP as a first nucleotide incorporated, which could be explained by the architecture of the RNAP active site (*Chamberlin and Ring, 1973*; *Kennedy et al., 2007*). Finally, CCPol-MP priming requires at least seven nt at the 3'end of the template (*Figure 8b*), potentially due to the natures of CCPol-MP binding, MP-ssDNA binding, and resulting spatial constraints.

Our experiments suggest that Cch2 may act as a self-loading helicase. It has ATP-dependent DNA helicase activity and binds specifically to a highly conserved sequence downstream from its coding sequence (*Figure 3b–d*), as do the SaPI replication initiators (*Ubeda et al., 2007*). However, other factors (for loading or activation) may be required, as we found only three iterons in the binding site, and our helicase activity in vitro was consistently low. It should be noted that for the founder of the D5 helicase family, from the vaccinia virus helicase-primase D5 protein, no helicase activity at all could be demonstrated in vitro, also suggesting that additional factors may be required (*Hutin et al., 2016*).

The data presented here, combined with our previous studies of pattern 1 SCC element-encoded proteins (*Mir-Sanchis et al., 2016*; *Mir-Sanchis et al., 2018*), strongly suggest that SCC elements are in fact replicative. A Cch or Cch2 operon can be found in all SCC elements identified to date, further supporting an essential role of the operon in SCC maintenance. Moreover, the Cch2 operon discussed here presents the same features as the replication modules of other replicative MGEs. For example, in nearly all DNA viruses that encode a replicative primase, a helicase domain is fused to the primase or is encoded in its vicinity (*Kazlauskas et al., 2016*). Furthermore, SaPI5 Rep, the closest homologue of Cch2, contains both primase and helicase domains, with putative SaPI origin of replication directly downstream from its gene (*Mir-Sanchis et al., 2016*; *Ubeda et al., 2007*). SCC replication remains to be demonstrated in vivo and it is not known what naturally activates expression of these genes. However, replication could serve several purposes: it could increase the stability of the element by providing a copy for reinsertion after excision, it provides a mechanism

for transient copy number expansion under selection, and it would greatly enhance the efficiency of any mechanism of horizontal transfer to new hosts.

It remains unclear whether additional SCC-, helper MGE-, or host-encoded factors are necessary for replication, and what the exact mechanism of SCC replication might be. Nevertheless, based on our biochemical analysis we can present a rough proposal for the initiation of pattern 2 SCC replication. Cch2 would bind to the origin of replication marked by the three iterons we identified, perhaps as two hexamers. After bubble opening and loading onto ssDNA by an unknown mechanism, Cch2 could then unwind the DNA, presenting a ssDNA template for priming by CCPol-MP, which shows a preference for and can efficiently begin priming at the sequences at the origin of replication. Since CCPol-MP does not appear to be highly processive and has no proofreading activity, a host polymerase would probably then use CCPol-MP-generated primers to efficiently complete replication of the element.

In summary, we have presented a biochemical characterization of three conserved proteins invariably encoded by pattern 2 SCC elements and likely involved in the replication of the element. Two of the proteins, CCPol and MP, form a complex acting as an intriguingly novel primase-polymerase, highlighting the importance of MGEs as reservoirs for proteins and protein complexes with potentially exciting enzymatic functions. Furthermore, since classical genetic approaches have been unsuccessful in elucidating SCC biology, we believe that this work and further characterizations of conserved SCC proteins will help advance our understanding of SCCs, most importantly of the SCC*mec* found in MRSA, the spread of which threatens new epidemics of multi-drug resistance *S. aureus* strains.

# Materials and methods

**Key resources table**

| Reagent type (species) or resource | Designation | Source or reference | Identifiers | Additional information |
|---|---|---|---|---|
| Gene (*Staphylococcus aureus*) | SCCmec | GenBank | AB512767.1 | Mobile genetic element |
| Gene (*Staphylococcus aureus*) | CCPol | GenBank | AB512767.1 ORF TS025; BAK57481.1 | |
| Gene (*Staphylococcus aureus*) | MP | GenBank | AB512767.1 ORF TS025; BAK57482.1 | |
| Gene (*Staphylococcus aureus*) | Cch2 | GenBank | AB512767.1 ORF TS025; BAK57483.1 | |
| Strain, strain background (*Escherichia coli*) | DH5alpha | Thermo Fischer | Cat # 18265017 | Cloning strain (chemically competent cells) |
| Strain, strain background (*Escherichia coli*) | BL21 (DE3) | Novagen (Sigma-Aldrich) | Cat # 69450 | Expression strain (chemically competent cells) |
| Strain, strain background (*Escherichia coli*) | Rosetta (DE3) pLysS | Novagen (Sigma-Aldrich) | Cat # 70956 | Expression strain (chemically competent cells) |
| Recombinant DNA reagent | pET21a (plasmid) | Novagen (Sigma-Aldrich) | Cat # 69740 | Expression plasmid |
| Recombinant DNA reagent | pET28a (plasmid) | Novagen (Sigma-Aldrich) | Cat # 69864 | Expression plasmid |
| Recombinant DNA reagent | pUC19 (plasmid) | NEB | Cat # N3041 | Cloning plasmid |

*Continued on next page*

*Continued*

| Reagent type (species) or resource | Designation | Source or reference | Identifiers | Additional information |
|---|---|---|---|---|
| Recombinant DNA reagent | pET28-SUMO | This work | | Expression plasmid with N-terminal SUMO tag introduced. Plasmid available upon reasonable request. |
| Recombinant DNA reagent | pET21-CCPol | This work | | CCPol expression plasmid (C-terminal His$_6$ tag). Plasmid available upon reasonable request. |
| Recombinant DNA reagent | pET28-Cch2 | This work | | Cch2 expression plasmid (TEV-cleavable N-terminal His$_6$ tag). Plasmid available upon reasonable request. |
| Recombinant DNA reagent | pET28-Cch2_operon | This work | | CCPol-MP-Cch2 operon expression plasmid (TEV-cleavable N-terminal His$_6$ tag on CCPol). Plasmid available upon reasonable request. |
| Recombinant DNA reagent | pET28-SUMO-CCPol+MP | This work | | CCPol-MP expression plasmid (terminal His$_6$-SUMO tag on CCPol). Plasmid available upon reasonable request. |
| Recombinant DNA reagent | pET28-SUMO-MP+CCPol | This work | | MP-CCPol expression plasmid (terminal His$_6$-SUMO tag on MP). Plasmid available upon reasonable request. |
| Recombinant DNA reagent | pET28-SUMO-CCPol+Cch2 | This work | | CCPol-Cch2 expression plasmid (terminal His$_6$-SUMO tag on CCPol). Plasmid available upon reasonable request. |
| Recombinant DNA reagent | pET28-SUMO-Cch2+CCPol | This work | | Cch2-CCPol expression plasmid (terminal His$_6$-SUMO tag on Cch2). Plasmid available upon reasonable request. |
| Recombinant DNA reagent | pET28-SUMO-Cch2+MP | This work | | Cch2-MP expression plasmid (terminal His$_6$-SUMO tag on Cch2). Plasmid available upon reasonable request. |
| Recombinant DNA reagent | pET28-SUMO-MP+Cch2 | This work | | MP-Cch2 expression plasmid (terminal His$_6$-SUMO tag on MP). Plasmid available upon reasonable request. |

*Continued on next page*

*Continued*

| Reagent type (species) or resource | Designation | Source or reference | Identifiers | Additional information |
|---|---|---|---|---|
| Recombinant DNA reagent | pET28-SUMO-MP+CCPol | This work | | MP-CCPol expression plasmid (no tags). Plasmid available upon reasonable request. |
| Recombinant DNA reagent | pET28-SUMO-CCPol+Cch2 | This work | | CCPol-Cch2 expression plasmid (no tags). Plasmid available upon reasonable request. |
| Recombinant DNA reagent | pET28-SUMO-MP+Cch2 | This work | | MP-Cch2 expression plasmid (no tags). Plasmid available upon reasonable request. |
| Recombinant DNA reagent | pUC19-ori | This work | | Plasmid containing the intergenic region between Cch2 and CcrC (putative origin of replication). Plasmid available upon reasonable request. |
| Sequence-based reagent | Synthesized oligonucleotides | IDT | Primers/oligonucleotides | For sequences, see *Supplementary file 1* |
| Commercial assay or kit | QuikChange Lightning Site-Directed Mutagenesis Kit | Agilent | Cat # 210519 | |
| Software, algorithm | ImageJ | NIH | https://imagej.nih.gov/ij/ | |
| Software, algorithm | Graphpad Prism | Graphpad Software, Inc, La Jolla, CA | http://www.graphpad.com/ | |

## DNA constructs

Type V SCC*mec* (GenBank: AB512767) genes and intergenic regions were cloned from genomic DNA of *S. aureus* TSGH17. CCPol (ORF TS025) was cloned into vector pET21a by sticky-end PCR (*Zeng, 1998*) with primers 1–4 (*Supplementary file 1*) using the NdeI and XhoI restriction sites to create the CCPol-6xHis fusion. Cch2 (ORF TS027) was cloned into vector pET28a with primers 5 and 6 (*Supplementary file 1*) using the BamHI and XhoI restriction sites. An N-terminal 6xHis tag followed by a thrombin cleavage site was introduced by restriction-free cloning with primers 7–8 (*Supplementary file 1*), then a TEV cleavage site was introduced using the QuikChange Lightning Site-Directed Mutagenesis Kit (Agilent) according to the manufacturer's instructions with primers 9–10 (*Supplementary file 1*) to create a 6xHis-TEV-Cch2 fusion construct with cleavable tag. This vector was then used to clone the intact Cch2 operon (ORFs TS025-TS027) by restriction-free cloning with primers 11 and 12 (*Supplementary file 1*), thus creating the 6xHis-TEV-CCPol + MP + Cch2 construct controlled by the T7 promoter. Dual expression constructs containing N-terminal hexahistidine and a Small Ubiquitin-like Modifier (SUMO) tags: 6xHis-SUMO-CCPol + MP, 6xHis-SUMO-MP + CCPol, 6xHis-SUMO-CCPol + Cch2, 6xHis-SUMO-Cch2 + CCPol, 6xHis-SUMO-Cch2 + MP, and 6xHis-SUMO-MP + Cch2 were created by polycistronic cloning with primers 5 and 13–17 (*Supplementary file 1*) into plasmid pET28-SUMO as described by EMBL Protein Expression and Purification Core Facility (www.embl.de/pepcore/pepcore_services/cloning/cloning_methods/ dicistronic_cloning), with a BamHI site in the forward and NheI/XhoI sites in the reverse primers. Control plasmids with constructs MP + CCPol, CCPol + Cch2, and MP + Cch2 were cloned with the same

method with primers 18–23 using restriction sites NcoI in the forward and NheI/BamHI in the reverse primer. CCPol D169A and D324A as well as Cch2 K252E were created by QuikChange mutagenesis with primers 24–29 (*Supplementary file 1*).

For the putative origin of replication, the whole intergenic region between the stop codon of Cch2 and the start codon of CcrC was amplified by PCR with primers 30–31 (*Supplementary file 1*) and inserted into vector pUC19 using BamHI and EcoRI restriction sites. All subsequent deletions within the putative ori region were created by QuikChange mutagenesis with primers 32–35 (*Supplementary file 1*). The random sequences with similar length and GC content as the putative ori region were amplified from the SenP2 protease gene (previously inserted into pET28a with NcoI and XhoI restriction sites) using primers 36–39 (*Supplementary file 1*). Their variants with inserted direct repeats were created by QuikChange mutagenesis with primers 40–43 (*Supplementary file 1*).

## Protein purification

All protein expression was carried out in Miller LB broth.

CCPol (with a C-terminal His$_6$ tag and a Leu-Glu linker between the natural C-terminus and the tag) was overexpressed in *E. coli* Rosetta (DE3) pLysS for 16 hr at 16˚C after addition of 0.5 mM iso-propyl β-D-1-thiogalactopyranoside (IPTG). Cells were lysed by sonication in purification buffer A (50 mM phosphate buffer, 5% glycerol, and 1 mM Dithiothreitol (DTT), pH 7.5) supplemented with cOmplete Protease Inhibitor Cocktail (Roche) and 200 μg/ml lysozyme. After centrifugation at 40,000 *g*, the pellet was resuspended in the purification buffer B (50 mM phosphate buffer, 1M NaCl, 5% glycerol, and 1 mM DTT, pH 7.5) and centrifuged again at 40,000. The protein was purified from the soluble fraction by Ni-affinity chromatography on a HisTrap column (GE Healthcare), eluted with purification buffer B supplemented with imidazole and stored in storage buffer A (20 mM Tris-HCl (pH 8.0), 0.5 mM EDTA, 1M NaCl, 20% Glycerol, 1 mM DTT, pH 8.0).

Cch2 and its mutated variant K252E were overexpressed from the 6xHis-TEV-Cch2 construct described above in *E. coli* BL21 (DE3) for 16 hr at 15˚C with 1 mM IPTG. Cells were lysed by sonication in purification buffer C (1x phosphate buffered saline (PBS), 1M NaCl, 5% glycerol, 15 mM imidazole, and 0.2 mM tris(2-carboxyethyl)phosphine (TCEP), pH 7.5) supplemented with cOmplete Protease Inhibitor Cocktail. The lysate was cleared by centrifugation at 40,000 *g*. The proteins were purified from the soluble fractions by Ni-affinity chromatography on a HisTrap column (eluted with purification buffer C supplemented with imidazole), followed by 6xHis tag cleavage with TEV protease, dialysis into purification buffer D (1x PBS, 1M NaCl, 5% glycerol, and 0.2 mM TCEP, pH 7.5), tag removal on a Ni-affinity column, and size exclusion chromatography (SEC) in purification buffer D on Superdex 200 column (GE Heathcare). The final protein contains one additional N-terminal Ser residue before the natural Met1.

The Cch2 operon proteins (6His-CCPol + MP + Cch2) were co-expressed in *E. coli* BL21 (DE3) for 16 hr at 16˚C with 1 mM IPTG. Cells were lysed by sonication in purification buffer E (1x PBS, 250 mM NaCl, 5% glycerol, 15 mM imidazole, and 0.2 mM TCEP, pH 7.5) supplemented with cOmplete Protease Inhibitor Cocktail. The lysate was cleared by centrifugation at 40,000 *g*. The proteins were co-purified from the soluble fraction by Ni-affinity chromatography on a HisTrap column and eluted in purification buffer E supplemented with imidazole followed by size exclusion chromatography (SEC) in purification buffer F (1x PBS, 250 mM NaCl, 5% glycerol, and 1 mM DTT, pH 7.5) on a Superdex 200 column (GE Heathcare).

The CCPol-MP complex and its variants were expressed from the 6xHis-SUMO-MP + CCPol construct described above and were overexpressed and purified similarly to Cch2, with SenP2 protease used instead of TEV, and with ion exchange chromatography on a Heparin column (GE Healthcare) as a final purification step replacing SEC, with binding in purification buffer G (1x PBS, 100 mM NaCl, 5% glycerol, and 1 mM DTT, pH 7.5) and elution in purification buffer H (1x PBS, 1M NaCl, 5% glycerol, and 1 mM DTT, pH 7.5). After cleavage by SenP2, MP contained one additional N-terminal Ser residue before the natural Met1.

Cch2, the Cch2 operon proteins, and CCPol-MP were concentrated on Amicon Ultra centrifugal filters (Merck Millipore) and stored in storage buffer B (20 mM Hepes-NaOH (pH 7.5), 200 mM NaCl, 5% Glycerol, 1 mM DTT, pH 7.5). For biochemical assays, all proteins were diluted in storage buffer B to required concentrations.

## Blue native polyacrylamide gel electrophoresis (BN-PAGE)

Purified Cch2 and CCPol-MP were analyzed on Blue Native (BN) PAGE. To 1 µl of a protein at approximately 7 mg/ml, 15 µl of BN-loading buffer (25 mM Hepes, 10% glycerol, 10 mM DTT, 0.5% Coomassie) was added. Samples were loaded onto a Novex WedgeWell 4–20% Tris-Glycine Gel (Invitrogen) and run first at 75 V for 30 mins and then at 150 V for further 3.5 hr in the running buffer (25 mM Tris-glycine, pH 7.5). The cathode buffer additionally contained 0.01% Coomassie Brilliant blue G-250 (Bio-Rad). Gels were then washed and stained again with Coomassie.

## Protein interaction assays

Protein pairs (6xHis-SUMO-CCPol + MP, 6xHis-SUMO-MP + CCPol, 6xHis-SUMO-CCPol + Cch2, 6xHis-SUMO-Cch2 + CCPol, 6xHis-SUMO-Cch2 + MP, and 6xHis-SUMO-MP + Cch2) were co-expressed in *E. coli* BL21 (DE3) for 16 hr at 15°C with 1 mM IPTG. The cells were harvested, resuspended in purification buffer E (1x PBS, 250 mM NaCl, 5% glycerol, 15 mM imidazole, and 0.2 mM TCEP, pH 7.5), and lysed by sonication. The lysate was cleared by centrifugation at 40,000 *g* and the supernatant applied onto Ni Sepharose beads (GE Healthcare). The samples were washed with purification buffer E supplemented with 50 mM imidazole and the proteins were eluted with the same buffer supplemented with 1M imidazole. The eluted fractions were analyzed by SDS-PAGE. As controls, constructs with two untagged proteins were analyzed in the same assay.

## Radioactive labeling of DNA substrates

All DNA oligonucleotides (purchased with standard desalting from IDT) were resuspended in water according to manufacturer's concentration estimates. $[\gamma\text{-}^{32}P]$-ATP used for 5'-end labeling of oligonucleotides was purchased from PerkinElmer. Two different labeling reactions were carried out. For experiments shown in *Figures 3e*, *4*, *5d*, *6*, *7* and *8*, the labeling reaction contained 1x T4 Polynucleotide Kinase Reaction Buffer, 10 units of T4 Polynucleotide Kinase (both from NEB), 2 µM oligonucleotide, and 10 µCi of $[\gamma\text{-}^{32}P]$-ATP in 10 µl reaction mix. The reactions were incubated at 37°C for 1 hr, brought up to 25 µl, and phenol-extracted. In order to remove all unincorporated $[\gamma\text{-}^{32}P]$-ATP, the samples were purified on Micro Bio-Spin 6 Columns (Bio-Rad) according to the manufacturer's instructions. Labeled oligonucleotides were eluted in a final volume of 50 µl. For experiments shown in *Figures 3c,d* and *5a,b*, where a more accurate knowledge of the DNA concentration was desired, the labeling reaction contained 1x T4 Polynucleotide Kinase Reaction Buffer, 10 units of T4 Polynucleotide Kinase (both from NEB), 0.85 µM oligonucleotide, and 15 µCi of $[\gamma\text{-}^{32}P]$-ATP in 30 µl reaction mix. The reactions were incubated at 37°C for 1 hr, phenol-extracted, and purified on a P6 column. At this point, 1 µl of the same oligo at 100 µM (as well as 1 µl of complementary oligo at 100 µM when dsDNA was to be obtained) was added to 10 µl of labeled oligo in a 20 µl volume, for the final DNA concentration of 5 µM. All concentrations of labeled oligonucleotides reported above are based on the labeling reaction input and do not account for potential sample loss during purification on the P6 column.

## DNA-binding assays

Protein-DNA binding was assessed by electrophoretic mobility shift assays (EMSA). To test binding of Cch2 to the putative origin of replication and its derivatives shown in *Figure 3b*, the DNA substrates were amplified by PCR from previously prepared plasmids using cloning primers (see *Supplementary file 1*), purified with QIAquick PCR purification kit (QIAGEN), and stored in water. Binding reactions (15 µl) included 25 nM of ds PCR-amplified DNA substrate and 1–5 µM protein as indicated in binding buffer A (20 mM Hepes-KOH pH 7.5, 5% glycerol, 0.5 mM EDTA, 1 mM DTT). The reactions were incubated at 37°C for 1 hr, and complex formation was assessed on native 6% TBE gels (Invitrogen) that were stained with SYBR Gold nucleic acid stain (Life Technologies) and visualized under UV light.

Binding of Cch2 and its isolated C-terminus (*Figure 3c,d*) to 23-nt ss- and ds-DNA oligonucleotides (*Supplementary file 1*) was tested in binding reactions (10 µl) including 5 nM 5'-$^{32}$P-labeled DNA and increasing protein concentrations (0, 10, 25, 50, 100, 200, 300, 400, and 500 nM) in binding buffer B (20 mM Hepes-KOH pH 7.5, 10 mM KCl, 5% glycerol, 0.5 mM EDTA, 1 mM DTT). Binding of CCPol-MP, CCPol D324A-MP, and CCPol (*Figure 5a,b*) to 30-nt ss- and ds-DNA oligonucleotides (*Supplementary file 1*) was tested in 10 µl binding reactions including 2 nM 5'-$^{32}$P-

labeled DNA and increasing protein concentrations (0, 2.5, 5, 10, 20, 40, 80, 120, and 200 nM) in binding buffer B. For CCPol-MP binding to 10–45 nt oligonucleotides (*Supplementary file 1*) shown in *Figure 5d*, binding reactions (15 µl) included 50 nM 5'-$^{32}$P-labeled DNA and 180 nM CCPol-MP in binding buffer A. All reactions were incubated at 37°C for 1 hr, and complex formation was assessed on native 8% polyacrylamide TBE gels. Gels were imaged using a Personal Molecular Imager (Bio-Rad) and, where applicable, images were quantified with ImageJ (*Abramoff et al., 2007*). The plot shown in *Figure 5c* was prepared in Prism 8.4.2 (GraphPad). Binding data were fit to the equation Y = Bmax*X/(Kd + X), where X = percent bound and Y = protein concentration. Bmax was constrained to 100 except for when fitting the Hill equation. For each experiment, the mean and standard deviation from two measurements for each point are plotted and the curves show the fit of the equation to the data.

## Helicase assays

Helicase assays were performed with 5'-$^{32}$P-labeled top-strand DNA oligonucleotides (*Supplementary file 1*) annealed to the appropriate bottom strand oligonucleotide for the desired duplex end structure (HELA-5F/HELA-6R: 6 bp mismatched; HELA-5F/HELA-8R: 6 bp 3' tail; HELA-9F/HELA-7R: 6 bp 5' tail; and HELA-9F/HELA-8: blunt-ended) as described in *Mir-Sanchis et al., 2016*. The reactions (20 µl) contained 20 nM 5'-$^{32}$P-labeled DNA substrate, 25 mM ATP (or another nucleotide as indicated), 25 µg/ml bovine serum albumin (BSA), and Cch2 (or its derivative) at indicated concentrations (1–3 µM) in 1x MES activity buffer (20 mM MES pH6, 20 mM MgCl$_2$, 5% glycerol, 0.5 mM EDTA, 1 mM DTT). The reactions were incubated at 37°C for 3 min, after which excess (0.5 µM) of the appropriate unlabeled top strand oligonucleotide was added. The reactions were then further incubated for 30 min at 37°C, after which they were stopped with the stop buffer (12 mM EDTA, 0.12% SDS, 6% glycerol, 0.05% bromophenol blue, 0.05% xylene cyanol). Reaction products were analyzed on native 12% polyacrylamide TBE gels imaged in the Personal Molecular Imager.

## Primer extension assays

Two types of primer extension assays were performed: (a) *Figure 4a*: CCPol or CCPol-MP was incubated with 0.67 µM primer-template duplex, 0.4 µCi [α-$^{32}$P]-dATP or -ATP, and 0.67 mM each dCTP, dGTP, and dTTP or CTP, GTP, and TTP in 1x HEPES activity buffer (20 mM Hepes-KOH pH 7.5, 5% glycerol, 20 mM MgCl$_2$, 0.5 mM EDTA, 1 mM DTT) for 30 min at 37°C. After this time, 0.67 mM unlabeled dATP/ATP was added and the reactions were incubated for further 30 min at 37°C to allow elongation of the products; or (b) *Figures 4b,d* and *6d* (right panel), and 7b: CCPol-MP and its derivatives were incubated with 25 nM 5'-$^{32}$P-labeled primer, 0.67 µM template, 0.67 mM each dATP, dCTP, dGTP, and dTTP in 1x HEPES activity buffer for 30 min at 37°C. All templates had a 3'terminal ddC (added using Terminal Transferase; NEB) or were 3'-phosphorylated (purchased from IDT) to prevent addition of nucleotides to the template. The metal dependence of extension assays in *Figure 7b* were performed in 1x HEPES activity buffer lacking MgCl$_2$, with 20 mM divalent metal ions added as indicated.

For both types of experiments, the reactions were stopped by addition of 20 µl 50 mM Ethylenediaminetetraacetic acid (EDTA) and the DNA was ethanol-precipitated in the presence of 0.1 µg/µl glycogen. Samples were run with loading buffer (1x TBE, 0.005% xylene cyanol, 0.005% bromophenol blue, and 90% formamide) on a 16% denaturing urea-TBE gel after heating at 98°C for 10 min. The gel was imaged in the Personal Molecular Imager. For the reverse transcriptase assay in *Figure 6d*, an RNA template (kindly provided by Professor Joseph Piccirilli at the University of Chicago) was used.

## Terminal deoxynucleotidyl transferase assay

Terminal transferase reactions (15 µl) were performed with 50 nM 5'-$^{32}$P-labeled DNA template T2 (*Supplementary file 1*), 0.67 mM dATP, dCTP, dGTP, dTTP, or a mixture of all of them, and 4.2 µM CCPol-MP or CCPol in 1x HEPES activity buffer. Reactions were incubated for 1 hr at 37°C, then stopped with 20 µl of 50 mM EDTA. The DNA was ethanol-precipitated in the presence of 0.1 µg/µl glycogen and the samples were run on a 16% denaturing urea-TBE gel in the loading buffer. The gel

was imaged in the Personal Molecular Imager. As a negative control, we added a 3'-ddC to the template with Terminal Transferase to prevent addition of any nucleotides.

### Primase assays

Two types of primase reactions were performed: (a) monitoring formation of template-product duplex on a non-denaturing gel (*Figure 6a,b*) and (b) monitoring priming product formation on a denaturing gel (*Figures 6c,d*, *7a,c,d* and *8a,d*).

a. Reactions (15 µl) were performed with 50 nM 5'-$^{32}$P-labeled template T2 (*Supplementary file 1*), 0.67 mM dNTP or NTP mixture, and 5 µM CCPol/CCPolA-MP or 5U Klenow (exo-) (NEB) in 1x HEPES activity buffer for 1 hr at 37°C. The template 3' end was blocked with ddC to prevent addition of nucleotides. As a control, 0.67 µM primer P1 was added as indicated. Reactions were stopped with the stop buffer (see Helicase assays) unless otherwise indicated. Some reactions were heated at 98°C for 5 min to denature the DNA duplex, with added excess (0.5 µM) of unlabeled template to trap the complementary strand after heating. The reactions were run on native 12% polyacrylamide TBE gels and imaged in the Personal Molecular Imager.

b. Reactions (15 µl) were performed with a template (*Supplementary file 1*), 0.8 µCi [α-$^{32}$P]-dATP, 0.67 mM each dCTP, dGTP, dTTP, and protein amounts as indicated in figure legends (1–5 µM or 5U for Klenow (exo-) polymerase) in 1x HEPES activity buffer. Template concentrations were as follows: ss oligonucleotide DNA: 0.67 µM and phage M13mp18 genomic DNA (NEB): 3.3 ng/µl. The template 3' ends were blocked with ddC or phosphorylation (as indicated) to prevent addition of nucleotides. As a control, 0.67 µM of primer (*Supplementary file 1*) was added as indicated. The metal dependence of priming assays in *Figure 7c* were performed in 1x HEPES activity buffer lacking MgCl$_2$, with 20 mM divalent metal ions added as indicated. Reactions were incubated for 30 min at 37°C. When desired, 0.67 mM unlabeled dATP was added and the reactions were incubated for further 30 min at 37°C to allow elongation of the products (since the concentration of labeled dATP in the priming reactions was the limiting factor for further elongation). The reactions were then stopped with 20 µl of 50 mM EDTA and the DNA was ethanol-precipitated. The samples in loading buffer were run on a 16% denaturing urea-TBE gel, and the gel was imaged in the Personal Molecular Imager.

## Acknowledgements

We thank Ying Pigli for preparation of the initial expression constructs, purification of CCPol and initial experiments confirming CCPol polymerase activity. We thank Professor Joseph Piccirilli for providing an RNA substrate for reverse transcriptase assays and Drs. Robert Daum and Susan Boyle-Vavra for the strain from which genomic DNA was made.

## Additional information

### Funding

| Funder | Grant reference number | Author |
| --- | --- | --- |
| National Institute of General Medical Sciences | R01 GM121655 | Aleksandra Bebel<br>Melissa A Walsh<br>Ignacio Mir-Sanchis<br>Phoebe A Rice |
| European Molecular Biology Organization | Long-Term Fellowship ALTF 65-2017 | Aleksandra Bebel |

The funders had no role in study design, data collection and interpretation, or the decision to submit the work for publication.

### Author contributions

Aleksandra Bebel, Funding acquisition, Investigation, Methodology, Writing - original draft; Melissa A Walsh, Investigation, performed the experiments with the Cch2 C-terminal domain; Ignacio Mir-

Sanchis, Conceptualization, identified the appropriate strain, purified *S. aureus* genomic DNA and made initial clones; Phoebe A Rice, Conceptualization, Supervision, Funding acquisition, Methodology, Writing - review and editing

### Author ORCIDs
Ignacio Mir-Sanchis (iD) https://orcid.org/0000-0002-6536-0045
Phoebe A Rice (iD) https://orcid.org/0000-0002-3467-341X

### Decision letter and Author response
Decision letter https://doi.org/10.7554/eLife.55478.sa1
Author response https://doi.org/10.7554/eLife.55478.sa2

## Additional files

### Supplementary files
• Supplementary file 1. Supplementary tables. Table 1. Primers used in the study. Table 2. Oligonucleotides used for in vitro assays. Table 3. Primase start sites identified in the study.

• Transparent reporting form

### Data availability
All data generated or analysed during this study are included in the manuscript and supporting files.

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
