## [Decision Letter]

Thank you for submitting your article "A novel DNA primase-helicase pair encoded by SCCmec elements" for consideration by *eLife*. Your article has been reviewed by three peer reviewers, one of whom is a member of our Board of Reviewing Editors, and the evaluation has been overseen by Cynthia Wolberger as the Senior Editor. The following individual involved in review of your submission has agreed to reveal their identity: Janice Pata (Reviewer #2).

The reviewers have discussed the reviews with one another and the Reviewing Editor has drafted this decision to help you prepare a revised submission.

Summary:

The consensus amongst the referees is that the revised paper should include more quantitative analysis of the data, in better support of the comparisons of activities that are made in the Discussion. In the absence of quantitation, the characterization of both the polymerase and primase activities seem incomplete. There was also a consensus that Figure 5 and its model were not sufficiently supported by the data. That said, the reviewers all agreed that this is a fascinating set of observations and they hope that you will be able to revise the work for eventual publication in *eLife*.

Please see the full reviews below for the requested revisions.

Reviewer #1:

In addition to its obvious medical significance, this is a fascinating system from which we can glean fundamentally new insights into the regulated modulation of polymerase function and diversification of helicase and primase behavior. The fact that helicase is sequence-specific is very unusual. The fact that MP can cause the polymerase to switch from being template dependent synthesizing primers *de novo* is interesting too. The unique primase characteristics are fascinating, representing a totally new paradigm (using dNTPs, having a unique fold). The combined function of the complex is interesting as well. The data indicate that SCC elements are fully replicative – which is important for understanding mechanisms for conferring resistance. Some specific suggestions and comments are below.

1) It is important to define "iteron" and perhaps other terms, as *eLife* is a general journal. Particularly because this paragraph is confusing overall (for example, the text jumps from a description of Cch2 as a hexameric protein to a description of the intergenic region, and the reader does not see the connection) it is important that the terms are clear.

2) Subsection “Cch2 binds a putative origin of replication downstream from its gene” – helicases with actual sequence specificity are really rare. Even RecC is only a degenerated helicase that can bind to Chi. Therefore, it would be good if the text mentioned this novelty and quantitatively described how much tighter the observed binding is relative to random sequence (see point below about the unnecessarily qualitative nature of the work).

3) Subsection “CCPol-MP has primase activity” – Figure 6. There is clearly primer-independent extension in the presence of MP – but it is a small amount relative to the case with primer, which is not really discussed. It would be helpful for the authors to provide an explanation.

4) From above, the apparent primer independent (primase) activity is really weak based on the data. To make the bold claim that this is a unique primase, this reviewer thinks that the products should be sequenced. Perhaps it would be easiest the old fashioned way, with chain terminators of each nucleotide to read out the sequence.

5) The following statement, for example, exemplifies one of the issues I have with the paper, "we found that MP confers tight ssDNA binding", however, the authors did not actually measure affinities so the statement is basically invalid. All the measurements in the paper are qualitative when it would have been a simple matter to conduct many of the experiments (such as the EMSAs) in a manner that would have yielded semi-quantitative Kd values (for example Figure 5), enabling accurate comparative statements in the paper.

6) Discussion, seventh paragraph: This section is contradictory. The experiments done at higher amounts of dATP do, in fact, contraindicate that the length is explained by limiting amounts of radiolabeled dATP. It is not clear why the authors are hedging so much. "We cannot exclude that the length of primers observed for CCPol-MP arose due to limiting amounts of radiolabelled dATP in the reactions (priming reactions were conducted with 9nM labelled dATP compared to the optimal 0.67 mM of dCTP, dTTP, and dGTP used), although doubling or quadrupling the amounts of labelled dATP in the priming reaction did not change the product size distribution, only increased the amount of primers produced (not shown).

Reviewer #2:

The authors describe the cloning, expression, purification and initial biochemical characterization of 3 proteins (CCPol, MP and Cch2) from a mobile genetic element (SCC*mec*) that carries a methicillin resistance gene in *Staphylococcus aureus*. These proteins are predicted, based on sequence conservation and gene organization, to be involved in the replication and propagation of SCC*mecs*, which are found in all methicillin-resistant *S. aureus* isolates.

The biochemical data presented in the manuscript convincingly demonstrate that:

– CCPol, MP and Cch2 form a soluble complex. (Figure 2)

– CCPol and MP form a heterodimeric complex. (Figure 2—figure supplement 1C)

– Cch2 forms large, oligomeric complexes, that migrate on a native protein gel at sizes consistent with hexamers and dodecamers. (Figure 2—figure supplement 1A)

– Cch2 binds to double-stranded DNA that contains three AAGTG directs repeats found in a 23-bp sequence downstream of the Cch2 gene in the SCC*mec* element – a sequence that is a putative origin of replication. (Figure 3A, B, C. Figure 3—figure supplement 1)

– The C-terminal 138 amino acids of Cch2 binds to the same ds-DNA fragment (Figure 3D)

– Cch2 displays helicase activity, with a preference for single-stranded DNA on the 3' side of the duplex region. (Figure 3E)

– CCPol and the CCPol-MP complex display DNA primer/template-dependent DNA polymerase extension activity, which is dependent on the two aspartates (D169 and D324) that are predicted to be magnesium-binding catalytic residues in CCPol, based on alignment with A-family DNA polymerases. (Figure 4A, B, D, Figure 4—figure supplements 1 and 2)

– The CCPol-MP complex and, to a much lesser extent, CCPol display 3' terminal transferase activity on single-stranded DNA, with a strong preference for addition of dCTP. (Figure 4C)

– MP confers strong ssDNA- and weak dsDNA-binding properties to the CCPol-MP complex, with DNA-containing one complex visible by EMSA on ssDNA of lengths 15 and 20, and at least 2 complexes on longer ssDNA. (Figure 5A, B, C)

– The CCPol-MP complex, but not CCPol alone, can initiate DNA synthesis de novo on ssDNA and the activity is dependent on the catalytic residues (D169 and D324) of CCPol. (Figure 6)

– The CCPol-MP complex can act as a reverse transcriptase on DNA-primed RNA, but could not initiated de novo primer synthesis. (Figure 6D)

– CCPol-MP preferentially initiates DNA primer synthesis at sites 8-14 nucleotides from the 3' end of the ssDNA template; a weak consensus sequence has intriguing similarity to the putative origin of replication sequence. (Figure 7)

Because of the importance of SSC*mec* elements and the lack of an understanding on how they move between cells, the findings reported are exciting, especially the novel observation that CCPol-MP has the ability to both synthesize DNA primers de novo and extend them in a template-dependent manner! These findings are all consistent with the idea presented that these proteins are involved in the propagation of the SSC*mec* elements.

Reviewer #3:

The manuscript identifies and then biochemically characterizes an ORF in a Pattern 2 SCC element that contains DNA polymerase, primase and helicase activities. Importantly the 3 gene products in this cluster physically associate and act at direct repeats to initiate DNA replication. This appears to be a functional replication module. Overall, the paper is in interesting and provide scientific value to this novel replication system, however, the experiments need better quantifications and are missing some importance facets to nail down the biochemistry of these complexes. Moreover, some of their conclusions are not backed up by the data.

1) The authors show an interaction between primase-helicase, helicase-pol and primase-pol (Figure 2 and Figure 2—figure supplement 2). They then focus their experiments primarily on the primase-pol interaction and activities. But does the DNA synthesis rate or processivity increase in the presence of helicase (pol-helicase interaction).

2) Subsection “CCPol and the CCPol-MP complexes are active polymerases”, second paragraph: They should be able to determine whether limiting dATP/ATP in the reaction or the template itself is limiting full length product. I see in Figure 4B on polyT that there is full length extension. So, they can either dope in 32pdATP into cold dNTPS to effectively maintain high nucleotide concentrations, or they should try this on different templates. It is possible that there is a template nucleotide that is not replicated effectively. This would be interesting. Or they need to eliminate 2nd structures with other sequences or polyN DNA templates.

3) Figure 3—figure supplement 2: can Cch2 helicase load onto DNA by itself? Or does it need nucleotides? Also what about loading onto circular DNA templates either ss or ds?

4) "roughly similar efficiency"- instead, they can quantify these results, may be calculate a binding constant. Same with Figure 3C, the curve for dsDNA binding by Cch2 should be fit to a binding equation. Lots of quantitative information is lost in this manuscript in favor of qualitative descriptions. This includes data in Figures 5 and 6.

5) Can the processivity of primer extension be compared for CCPol-MP vs. CCPol? I would expect MP to increase processivity. A single-turnover assay should be performed.

6) Figure 5C and subsection “MP contributes ssDNA-binding activity to the CCPol-MP complex”: while it is true that the increasing substrate length should decrease the mobility, can the 5nt difference in the protein-DNA be seen in this small EMSA? The increase in mobility for the 15 to 30 nt bound states is not apparent on this gel as stated by the authors

7) Figure 5C. Each CCPol-MP needs at least 15 nts to load onto the DNA. They see the first DNA-protein complex oligomer formation around 15-20nts. And that oligomer state remains the same at 25nt. Only at 30nt (15+15) they see the second higher oligomer state. The next higher oligomer state should be seen at around 45nt (30+15). I don't see any evidence for an unusual shift in mobility that can be ascribed to DNA bending. Further experiments are needed to confirm bending and wrapping.

8) Alternatively, SEC can be performed with different length DNA to show a shift to multiple bound CCPol-MP based on the 15 base site size.

9) Discussion, fourth paragraph, metal dependence. It would be interesting and important to show the metal dependence for priming, Mg, Mn, Co, Zn, etc….

10) Seems like the primase starting site is 7 bases from the 3' end of the primer in these experiments, but this is not really biologically relevant. What about priming on bubble substrates or blocking the 3' end with biotin/SA or something else to rule out 3' specific binding.

11) Figure 7, As there does not appear to be a consistent primer initiation site, besides starting with a purine G, is there a consistent primer length or again relating back to point 3, is the major pause/stop site sequence dependent.

12) In the Discussion they talk about their hypothesis that coupled origin melting, helicase activation, and then template priming is important for beginning replication, however there are no coupled assays to show this. It would be really exciting to see that, however, I do note that the authors state that unwinding is limiting, so that may not be possible here. What about on a bubble substrate that mimics the DR? Can the combination of all three proteins stimulate unwinding? Or priming? Or polymerization (processivity)?

---

## [Author Response]

Reviewer #1:[…] Some specific suggestions and comments are below.1) It is important to define "iteron" and perhaps other terms, as eLife is a general journal. Particularly because this paragraph is confusing overall (for example, the text jumps from a description of cch2 as a hexameric protein to a description of the intergenic region, and the reader does not see the connection) it is important that the terms are clear.

First, we changed the subheading to “Cch2 forms oligomers and binds specifically to a putative origin of replication” to make the topic switch between paragraphs less jarring.

Second, the paragraph in question has been rewritten to better explain iterons and our logic in general.

2) Subsection “Cch2 binds a putative origin of replication downstream from its gene” – helicases with actual sequence specificity are really rare. Even RecC is only a degenerated helicase that can bind to Chi. Therefore, it would be good if the text mentioned this novelty and quantitatively described how much tighter the observed binding is relative to random sequence (see point below about the unnecessarily qualitative nature of the work).

The Introduction now includes a note about analogous viral helicases that recognize their own origins of replication in dsDNA – note that these (like Cch2) recognize sequences in double-stranded DNA whereas RecC recognizes single stranded chi.

Also, the new binding assays were done under more quantitative conditions (careful titrations with [DNA] < Kd). Unfortunately, for Cch2 (but not for CCPol-MP) this led to smearing and material in the wells that made accurate estimations difficult.

3) Subsection “CCPol-MP has primase activity” – Figure 6. There is clearly primer-independent extension in the presence of MP – but it is a small amount relative to the case with primer, which is not really discussed. It would be helpful for the authors to provide an explanation.

We suspect that initiation of DNA synthesis is less efficient than simple elongation – the rate limiting step may be binding the first two dNTP’s and/or making that first bond between them. We have added a note in the main text to this effect.

4) From above, the apparent primer independent (primase) activity is really weak based on the data. To make the bold claim that this is a unique primase, this reviewer thinks that the products should be sequenced Perhaps it would be easiest the old fashioned way, with chain terminators of each nucleotide to read out the sequence.

Unfortunately, we ran out of time to do this – because the initiation site is variable it is possible but not quite as simple as hoped. However, the other assays clearly show primase activity, and noted above, one might expect *de novo* initiation to be more difficult than extension.

5) The following statement, for example, exemplifies one of the issues I have with the paper, "we found that MP confers tight ssDNA binding", however, the authors did not actually measure affinities so the statement is basically invalid. All the measurements in the paper are qualitative when it would have been a simple matter to conduct many of the experiments (such as the EMSAs) in a manner that would have yielded semi-quantitative Kd values (for example Figure 5), enabling accurate comparative statements in the paper.

The binding assays in question have been all repeated in a more quantitative manner resulting in an almost entirely new Figure 5 and calculations of the apparent Kd and Hill coefficient.

6) Discussion, seventh paragraph: This section is contradictory. The experiments done at higher amounts of dATP do, in fact, contraindicate that the length is explained by limiting amounts of radiolabeled dATP. It is not clear why the authors are hedging so much. "We cannot exclude that the length of primers observed for CCPol-MP arose due to limiting amounts of radiolabelled dATP in the reactions (priming reactions were conducted with 9nM labelled dATP compared to the optimal 0.67 mM of dCTP, dTTP, and dGTP used), although doubling or quadrupling the amounts of labelled dATP in the priming reaction did not change the product size distribution, only increased the amount of primers produced (not shown).

We realized this was confusing and have re-written that paragraph.

Our original point was that even though 4 x 9 = 36 nM is still far less than 0.67mM, if limiting dATP was the only thing limiting product size, we might have seen at least some increase in primer length rather than simply an increase in primer amount when quadrupling the labeled dATP concentration.

We suspect that very high dNTP concentrations somehow kick our system over an intrinsic barrier to elongation past ~15nt. However, we agree that the experiments presented aren’t enough to really nail down the conclusion. Therefore, we have simplified the Discussion by referring to what is now Figure 8 (old 7) where the concentration of dATP was raised to .67 mM (to match that of the other dNTPs) and the primers were extended to full length.

Reviewer #3:(…)1) The authors show an interaction between primase-helicase, helicase-pol and primase-pol (Figure 2 and Figure 2—figure supplement 2). They then focus their experiments primarily on the primase-pol interaction and activities. But does the DNA synthesis rate or processivity increase in the presence of helicase (pol-helicase interaction).

This is an interesting question but because the helicase activity is weak and dependent on a specific sequence in the dsDNA part of the substrate for reasons that we can’t fully explain mechanistically, we think it is beyond the scope of the study at this point.

2) Subsection “CCPol and the CCPol-MP complexes are active polymerases”, second paragraph: They should be able to determine whether limiting dATP/ATP in the reaction or the template itself is limiting full length product. I see in Figure 4B on polyT that there is full length extension. So, they can either dope in 32pdATP into cold dNTPS to effectively maintain high nucleotide concentrations, or they should try this on different templates. It is possible that there is a template nucleotide that is not replicated effectively. This would be interesting. Or they need to eliminate 2nd structures with other sequences or polyN DNA templates.

We repeated Figure 4A adding additional cold dATP after a long enough incubation to get a hot enough product to see. From this we can conclude that it isn’t the template itself – it was the limiting dATP.

There is a misunderstanding about Figure 4B: The “T” in T1, T2, T3 and T4 refers to “Template” not “Thymidine” – these templates are not poly-dT. To clarify we have inserted a note in both the main text and figure legend pointing to Supplementary file 1—supplementary table 2 where the sequences of these templates can be found.

3) Figure 3— figure supplement 2: can Cch2 helicase load onto DNA by itself? Or does it need nucleotides? Also what about loading onto circular DNA templates either ss or ds?

We suspect that we didn’t see binding to linear ssDNA because it can probably slide off the ends, and have added a note in the text saying so. Small circular DNAs are a good idea but we did not have time to try this before the lab closed. Regarding circular dsDNA: because the specific dsDNA sequence is needed for helicase activity, we have not yet been able to distinguish unambiguously between the binding mode that is most likely occurring in Figure 3 where the C-terminal domain is bound to dsDNA versus a more canonical helicase-like mode with the DNA through the hole in the protein ring. This will hopefully be more straightforward in the future once we have figured out what is limiting our helicase activity, but that may require substantial additional work. For this manuscript we feel that the main feature is the new primase activity, with some characterization of the helicase that it interacts with as a “bonus”.

4) "roughly similar efficiency"- instead, they can quantify these results, may be calculate a binding constant. Same with Figure 3C, the curve for dsDNA binding by Cch2 should be fit to a binding equation. Lots of quantitative information is lost in this manuscript in favor of qualitative descriptions. This includes data in Figures 5 and 6.

Thank you for the impetus to improve this paper. As noted above, the binding assays have now been redone under more quantitative conditions. Unfortunately, for Cch2 binding the resulting gels are not rigorously quantifiable because those conditions led to some smearing and material in the well in addition to a discrete shifted band. Although with more time we might have been able to perfect conditions for these binding assays, we feel that these new gels, in combination with the earlier assays on longer DNAs that don’t yield Kd but show two discrete shifted bands, are sufficient characterization to make the points that are most important for this paper.

However, please note that the new set of CCPol-MP binding assays has been fit to binding curves now as requested, and the section of text has been completely rewritten.

5) Can the processivity of primer extension be compared for CCPol-MP vs. CCPol? I would expect MP to increase processivity. A single-turnover assay should be performed.

We attempted this assay just before the lab closed down and sadly the answer was that we need to do a lot more work adjusting the conditions before we can make reliable, printable conclusions. Although we agree that this would add to the paper we don’t think it’s critical to our central points.

6) Figure 5C and subsection “MP contributes ssDNA-binding activity to the CCPol-MP complex”: while it is true that the increasing substrate length should decrease the mobility, can the 5nt difference in the protein-DNA be seen in this small EMSA? The increase in mobility for the 15 to 30 nt bound states is not apparent on this gel as stated by the authors

Close examination shows a very subtle increase in mobility and certainly no decrease as one might expect for longer DNAs. However, the increase is so subtle and the point is so peripheral to our main story that we simply deleted this as noted above.

7) Figure 5C. Each CCPol-MP needs at least 15 nts to load onto the DNA. They see the first DNA-protein complex oligomer formation around 15-20nts. And that oligomer state remains the same at 25nt. Only at 30nt (15+15) they see the second higher oligomer state. The next higher oligomer state should be seen at around 45nt (30+15). I don't see any evidence for an unusual shift in mobility that can be ascribed to DNA bending. Further experiments are needed to confirm bending and wrapping.

See answer #6 above: offending speculation deleted.

8) Alternatively, SEC can be performed with different length DNA to show a shift to multiple bound CCPol-MP based on the 15 base site size.

Theoretically it could but as that would be complementary if not somewhat redundant to the gel-based binding assays above and our lab doesn’t own the appropriate analytical sizing column, that experiment was lower on our to-do list and we didn’t have time for it.

9) Discussion, fourth paragraph, metal dependence. It would be interesting and important to show the metal dependence for priming, Mg, Mn, Co, Zn, etc….

Completed as requested – these experiments are now the new Figure 7B and C.

10) Seems like the primase starting site is 7 bases from the 3' end of the primer in these experiments, but this is not really biologically relevant. What about priming on bubble substrates or blocking the 3' end with biotin/SA or something else to rule out 3' specific binding.

Initially, we were also curious as to whether or not there was any specific binding to a 3’ end. However, we suspect not for three reasons. First, priming was successful whether the template’s 3’ end was blocked with a dideoxyC (Figure 6A, B) or a phosphate (Figure 6C, D and 7A, D (=old 6E, F)). Second, in the Figure 6C panel 2, priming products were seen on a circular M13 template. Third, in Figure 8 (old 7) the distance from the priming site to the three prime end is variable. We have added a note in the results for Figure 6C noting this.

11) Figure 7, As there does not appear to be a consistent primer initiation site, besides starting with a purine G, is there a consistent primer length or again relating back to point 3, is the major pause/stop site sequence dependent.

First please note our answer above hopefully clearing up the misunderstanding in point 3.

Second, we have edited the Results and Discussion to clear up some confusion resulting from the fact that Figures 6C, D and 7A, D (old 6C-F) were done using a limiting concentration of radiolabeled dATP (to drive incorporation of enough label to see the products) whereas in Figure 8 (old 7) we added additional cold dATP later in the reaction which results in extension of those primers, as seen in previous primer extension assays that used synthetic primers.

We are intrigued that when dATP is limiting we see a roughly consistent primer size made even on different templates and would like to follow this up later with a much more extensive and detailed characterization of the enzymology of the CCPol-MP complex. However, we feel that is beyond the scope of this paper.

12) In the Discussion they talk about their hypothesis that coupled origin melting, helicase activation, and then template priming is important for beginning replication, however there are no coupled assays to show this. It would be really exciting to see that, however, I do note that the authors state that unwinding is limiting, so that may not be possible here. What about on a bubble substrate that mimics the DR? Can the combination of all three proteins stimulate unwinding? Or priming? Or polymerization (processivity)?

The reviewer is correct that due to the ornery nature of this helicase (which probably reflects something we don’t know about it yet) our attempts at coupled assays on a variety of substrates have so far been ambiguous.